# The CRISPR-Cas13a Gemini System for noncontiguous target RNA activation

Hongrui Zhao[1,2,4], Yan Sheng [2,3,4] ✉, Tenghua Zhang[2], Shujun Zhou[2], Yuqing Zhu[2], Feiyang Qian[2], Meiru Liu[1], Weixue Xu[1], Dengsong Zhang [1] ✉ & Jiaming Hu [1,2] ✉

Simultaneous multi-target detection and multi-site gene editing are two key factors restricting the development of disease diagnostic and treatment technologies. Despite numerous explorations on the source, classification, functional features, crystal structure, applications and engineering of CRISPR-Cas13a, all reports use the contiguous target RNA activation paradigm that only enables single-target detection in vitro and one-site gene editing in vivo. Here we propose a noncontiguous target RNA activation paradigm of Cas13a and establish a CRISPR-Cas13a Gemini System composed of two Cas13a:crRNA binary complexes, which can provide rapid, simultaneous, highly specific and sensitive detection of two RNAs in a single readout, as well as parallel dual transgene knockdown. CRISPR-Cas13a Gemini System are demonstrated in the detection of two miRNAs (miR-155 and miR-375) for breast cancer diagnosis and two small RNAs (EBER-1 and EBER-2) for Epstein-Barr virus diagnosis using multiple diagnostic platforms, including fluorescence and colorimetric-based lateral flow systems. We also show that CRISPR-Cas13a Gemini System can knockdown two foreign genes (EGFP and mCherry transcripts) in mammalian cells simultaneously. These findings suggest the potential of highly effective and simultaneous detection of multiple biomarkers and gene editing of multiple sites.

The signature effector protein of class 2 type VI-A CRISPR-Cas systems[1–5], Cas13a (formerly C2c2), is an RNA-guided and RNA-activated RNA endonuclease (RNase) that is complexed with a precursor CRISPR RNA (pre-crRNA) containing a repeat region and a programmable guide region and then cleaves pre-crRNA within the repeat region to generate mature crRNA and remains bound after cleavage, forming a nuclease-inactive ribonucleoprotein complex (Cas13a:crRNA binary complex). The formed Cas13a:crRNA binary complex binds to the complementary target RNA (also referred to RNA activator) which consequently activates the higher eukaryotes and

prokaryotes nucleotide-binding (HEPN) domains of Cas13a for RNA cleavage[6,7]. Based on its natural RNase activity against target RNAs, together with collateral cleavage of nonspecific RNAs, Cas13a has been recently used for RNA manipulation in eukaryotic cells and nucleic acid detection in vitro[6–18].

Furthermore, among all the Cas13a homologs, *Leptotrichia buccalis* (Lbu) Cas13a exhibits the superior cleavage activity and prefers homo-uridine as the substrate for trans-RNA cleavage[10]. Wang and her colleagues have conducted molecular architecture and mechanism analysis of LbuCas13a nuclease and its complexes. They determined

[1]International Joint Laboratory of Catalytic Chemistry, State Key Laboratory of Advanced Special Steel, Innovation Institute of Carbon Neutrality, College of Sciences, Shanghai University, Shanghai, China. [2]MOE Key Laboratory of Laser Life Science & Institute of Laser Life Science, Guangdong Provincial Key Laboratory of Laser Life Science, College of Biophotonics, South China Normal University, Guangzhou, China. [3]Institute of Translational Medicine, Shanghai University, Shanghai, China. [4]These authors contributed equally: Hongrui Zhao, Yan Sheng. ✉e-mail: ysheng@shu.edu.cn; dszhang@shu.edu.cn; jmhu@shu.edu.cn

the crystal structure of LbuCas13a bound to crRNA and its target RNA, as well as the cryo-EM structure of the LbuCas13a:crRNA binary complex and revealed the mechanism of target RNA recognition and RNA cleavage by LbuCas13a[19,20]. Moreover, it has been shown that the binding affinity of target RNA to Cas13a:crRNA and the HEPN-nuclease activity of Cas13a are decoupled and differentially affected by the mismatches between crRNA and target RNA[21].

Despite numerous explorations on the mismatch and length of target RNA, all reported studies used the contiguous target RNA activation paradigm, which could only achieve one-site gene editing and single-target detection. To the best of our knowledge, there has been no report on the noncontiguous target RNA activation paradigm of CRISPR-Cas13a. The importance of exploring the noncontiguous target RNA activation paradigm lies in breaking through the existing knowledge of Cas13a nuclease activation and looking for the possibility of gene editing of two sites and dual nucleic acid detection, with the aim of having a positive signal only when both targets are present.

In this work, we use LbuCas13a in complex with mature crRNA and 11 groups of target RNAs to systematically explore the possibility and summarize the regularity of noncontiguous target RNA activation of CRISPR-Cas13a. And based on this feature, we further construct a system in which two target RNAs activate one Cas13a nuclease with assistance of an auxiliary sequence, and finally establish the CRISPR-Cas13a Gemini System: two independent target RNAs co-activate dual Cas13a nucleases. We demonstrate the universal nature of CRISPR-Cas13a Gemini System through its utility in transgene knockdown of two sites (EGFP and mCherry transcripts) and dual miRNAs or small RNAs detection in a single readout for early cancer or virion diagnosis on various detection platforms, including fluorescence and colorimetric lateral flow assay (LFA) systems.

## Results

### Noncontiguous target RNA activation paradigm of Cas13a

To systematically explore the noncontiguous target RNA activation paradigm of Cas13a, two strategies were developed (Fig. 1a). Strategy I consists of two short RNA sequences, which together are fully complementary to the guide region of crRNA. Whereas strategy II is a long RNA sequence, 5′ and 3′ ends of which together are completely complementary to the guide region of crRNA, forming a loop constructure. The 50-nt crRNA sequence comprises a 30-nt repeat region (brown) and a 20-nt guide region (red) (Fig. 1b). Based on the binding affinity between Cas13a:crRNA binary complex and target RNA, 11 break sites (site 1–11) were designed from nucleotide 5 to 15 in the guide region, and 11 groups ($G_{5-15}$–$G_{15-5}$) of target RNAs were developed accordingly (Fig. 1b and Supplementary Table 1). Each group consisted of strategy I and strategy II with the same break site, respectively.

Considering the instability of base complementarity between short sequences and the importance of the central seed region within the guide strand[6,19,21], $G_{7-13}$ containing Target 7 and Target 13′ ($T_7 + T_{13'}$) of strategy I and Target 7 + 13′ ($T_{7+13'}$) of strategy II was first chosen to verify the noncontiguous target RNA activation of Cas13a (Fig. 1c). A fluorescent RNA cleavage assay was used to assess HEPN-mediated RNA cleavage upon RNA mediated Cas13a activation. The data revealed the disability of $T_7 + T_{13'}$ to activate Cas13a trans-RNA cleavage, while $T_{7+13'}$ was able to promote robust Cas13a HEPN-nuclease activity, with cleavage rate close to the contiguous target RNA ($T_{20}$) (Fig. 1d). We speculate that the difference in the results of the two activation strategies are probably due to the proximity effects of nucleic acid strands[22–25]. Specifically, when 13 nucleotides at the 3′ end of $T_{7+13'}$ are complementary to the crRNA, the binding probability of the 7 nucleotides at its 5′ end to crRNA will be greatly increased, thereby activating Cas13a.

Representative time course of fluorescence measurements was generated by Cas13a HEPN-nuclease activation by the addition of $G_{5-15}$ to $G_{15-5}$ (Supplementary Fig. 1a, b). Apparent cleavage rates

relative to contiguous target RNA ($T_{20}$) were determined from each of the resulting time courses, and comparative analysis suggests that activation of Cas13a by strategy II forming a loop constructure generated sufficient and detectable fluorescence signals and showed a "M"-shaped wave trend (Fig. 1e). Whereas strategy I were less capable of activating Cas13a HEPN-nuclease activity, and even failed. Notably, the $T_{9+11'}$ resulted in more robust Cas13a HEPN-nuclease activation relative to others, which was further verified by another set of crRNA and target RNA (noncontiguous at site 9) (Supplementary Fig. 2). $T_{7+13'}$, $T_{8+12'}$, $T_{11+9'}$, $T_{12+8'}$, and $T_{13+7'}$ had the second highest ability to activate Cas13a. More surprisingly, $T_{10+10'}$ was not able to efficiently trigger the Cas13a HEPN-nuclease activity, even though 10 nucleotides ($\Delta G = -29.23$ kcal/mol) in $T_{10+10'}$ was bound more tightly by Cas13a:crRNA than 8 ($\Delta G = -25.74$ kcal/mol) or 9 nucleotides ($\Delta G = -26.41$ kcal/mol), simulated and calculated by NUPACK. We concluded that Cas13a target RNA binding affinity and nuclease activation are decoupled and differentially affected by break sites between the target RNA and the guide region of crRNA. The low efficiency in Cas13a catalytic activation exhibited by $T_{5+15'}$, $T_{6+14'}$, $T_{14+6'}$, and $T_{15+5'}$ could be attributed to their weak binding affinities with Cas13a:crRNA binary complex that are unable to form a base-paired RNA duplex in the important regions for HEPN-nuclease activation. Furthermore, it is notable that $T_{14} + T_{6'}$ and $T_{15} + T_{5'}$ exhibited abilities to activate HEPN-nuclease activity of Cas13a, with even $T_{15} + T_{5'}$ activating Cas13a more efficiently than $T_{15+5'}$. We reasoned that even though the binding of the 15 nucleotides at the 5′ end of $T_{15+5'}$ to the guide strand of crRNA has a certain proximity effect on the 3′ end of $T_{15+5'}$, due to the weak binding affinities between the 5 nucleotides at the 3′ end of $T_{15+5'}$ and the guide strand of crRNA, $T_{15+5'}$ may not form a loop constructure with the crRNA. Compared with $T_{15}$ alone, there is a long extension at the 3′ end of crRNA-$T_{15+5'}$ duplex, which might weaken the Cas13a nuclease activation. To demonstrate this hypothesis, we established two control groups that a long RNA sequence contains varied complementary sequence (5–15-nt) to the guide strand of crRNA at its 5′ end and non-targeted sequence at its 3′ end ($T_{n+non'}$), and vice versa ($T_{non+n'}$) (Supplementary Fig. 3a and Supplementary Table 2). Since these two control groups possess only single crRNA targeting region, we compared their efficiencies of non-target fluorescent RNA cleavage to strategy I (Supplementary Fig. 3b). Comparative analysis reveals that $T_{non+n'}$ was unable to activate Cas13a for collateral RNA cleavage. $T_{n+non'}$ with no more than 12-nt complementary sequence from its 5′ end failed to activate HEPN-nuclease activity of Cas13a, while $T_{n+non'}$ with 13–15-nt complementary sequence from its 5′ end exhibits abilities for activating Cas13a, but with lower cleavage rate compared to strategy I. However, the previous studies reported that a 20-bp guide-RNA duplex was essential for activating Cas13a and target RNA shorter than 16-nt in length failed to trigger Cas13a HEPN-nuclease activity[7,19]. To further investigate the shortest length requirement of the target RNA for activation of Cas13a HEPN-nuclease, we measured the efficiency of Cas13a trans-RNA cleavage in the presence of crRNA and complementary target RNAs of 10–15-nt in length (Fig. 1f), and found that target RNA with 15- or even 14-nt complementary sequence from its 3′ end was still able to promote the HEPN-nuclease activity of Cas13a, while target RNAs shorter than 14-nt showed no cleavage. Besides, target RNA shortened by 5-nt or more from its 3′ end ($T_{15'}$ to $T_{10'}$) was also unable to activate Cas13a for collateral RNA cleavage. Electrophoretic gel analysis further verified this phenomenon (Fig. 1g and Supplementary Fig. 4). The possible reason is that the designed crRNA sequence with a missing base at the 3′ end of its repeat region (crRNA#) in this work is slightly different from the reported sequence with adenine base at the 3′ end of its repeat region (crRNA-A)[7,10]. To investigate the influence of the base at the 3′ end of the crRNA repeat region on Cas13a activity, we compared 4 crRNAs, including crRNAs with 4 different bases at the 3′ end of the repeat region (crRNA-A,

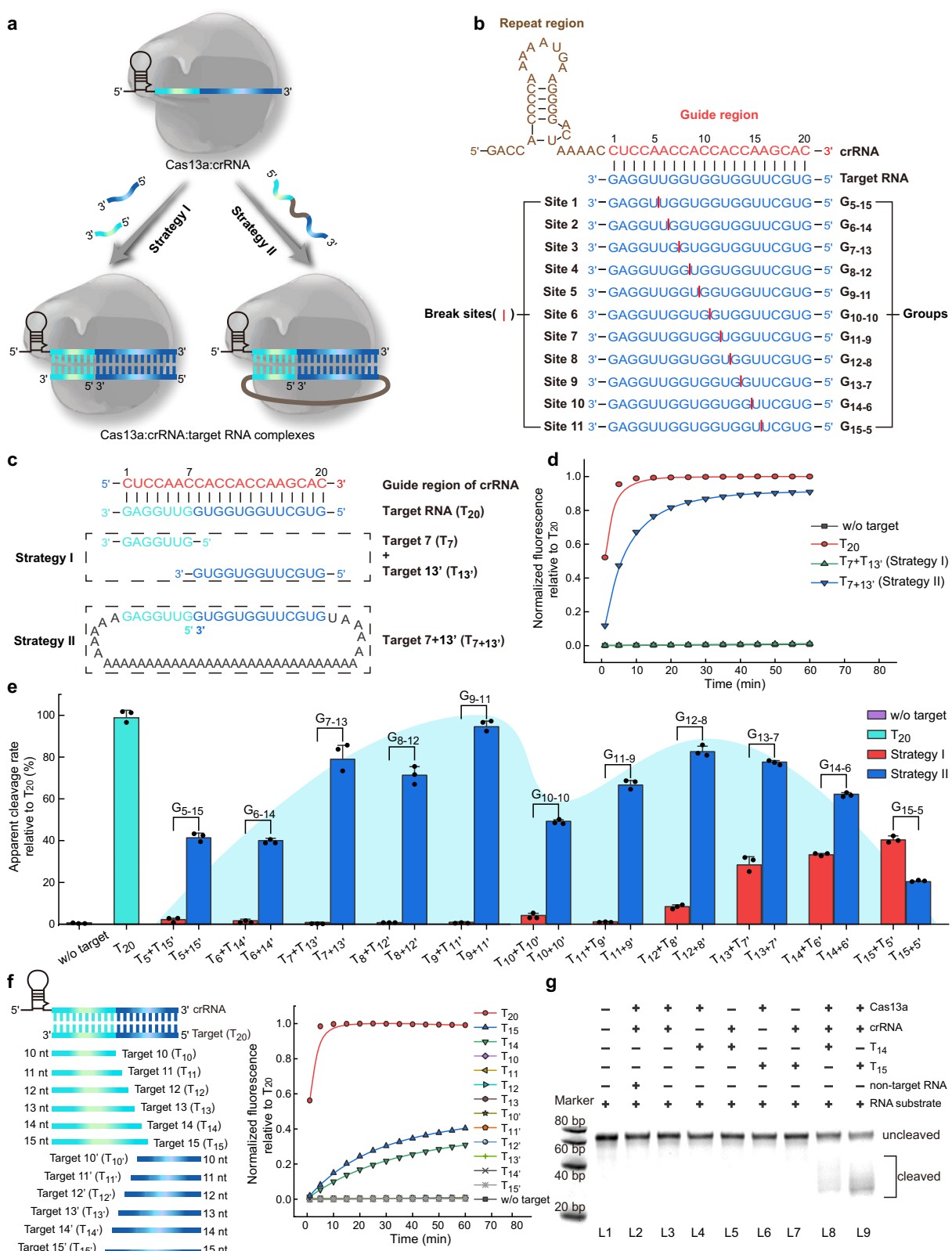

crRNA-U, crRNA-C and crRNA-G), with crRNA# (Supplementary Fig. 5). The data show that $T_{14}$ and $T_{15}$ could effectively activate Cas13a after binding with crRNA#. By contrast, crRNA-C group had a weak signal, and neither $T_{14}$ nor $T_{15}$ could activate Cas13a in the presence of crRNA-A, crRNA-U or crRNA-G. These results update the shortest length requirement of target RNA for activating the CRISPR-Cas13a system.

Besides the strategy I and II, three additional models have been designed to test the generalizability of our method (Supplementary Fig. 6). Comparison experiments between short RNA sequences (model 1A, 2A, and 3A) and long RNA sequences (model 1B, 2B, and 3B) with different locations of target region were performed using $G_{9-11}$. The results show that short RNA sequences in these three models exhibited no Cas13a trans-RNA cleavage regardless of the non-target

**Fig. 1 | Noncontiguous target RNA activation paradigm of Cas13a. a** Schematic of two strategies for activating Cas13a. **b** Schematic of 11 break sites (site 1–11) on guide region of crRNA (nucleotides 5–15), and corresponding 11 groups of target RNAs ($G_{5-15}$–$G_{15-5}$). The repeat and guide region of crRNA are shown in brown and red, respectively and the target RNAs are shown in blue. The break sites are indicated as red lines. **c** Sequences of strategy I and strategy II of $G_{7-13}$. **d** Fluorescence analysis of Cas13a HEPN-nuclease activation by $G_{7-13}$. w/o is the abbreviation of without. **e** Apparent cleavage rate of 11 groups ($G_{5-15}$–$G_{15-5}$) of target RNAs-mediated Cas13a trans-cleavage reporter relative to contiguous target ($T_{20}$). w/o target and

$T_{20}$ shown as negative and positive controls, and the red and blue bars corresponded to the activation mode of strategy I and strategy II, respectively. Blue shading represents the "M"-shaped wave trend. Data are presented as mean values ± standard deviation from three independent experiments. **f** Fluorescence analysis of 10–15-nt target RNA activation of Cas13a. $T_{14}$ (green) and $T_{15}$ (blue) can activate the HEPN-nuclease activity of Cas13a to some extent. **g** Electrophoretic gel analysis of Cas13a activation by $T_{14}$ or $T_{15}$. The experiment was repeated twice independently with similar results. Source data are provided as a Source data file.

extension sequences at the 5′ and/or 3′ end, but Cas13a HEPN-nuclease activity could be triggered when forming a loop constructure with the long RNA sequence regardless of whether the target region is at the end or other locations of the sequence. These additional models demonstrate the spacer sequence of crRNA is not restricted to the 5′ and 3′ end of any RNA targets, indicating the broad generalizability of our method.

## Two RNA activators co-activate Cas13a with the assistance of an auxiliary sequence

Next, we wondered whether Cas13a could be activated for trans-RNA cleavage as two target RNAs forming a loop constructure with the crRNA. As $T_{9+11'}$ promotes the highest Cas13a HEPN-nuclease catalytic activity, it was selected for the subsequent experiments. We designed two RNA activators, consisting of three functional domains: crRNA binding site (cerulean or blue), bumper domain (gray) and bridge binding site (black), and an auxiliary DNA strand as a bridge to connect the two RNA activators by hybridizing with their bridge binding sites, respectively (Fig. 2a). Specifically, the 9 nucleotides at the 5′ end of the RNA activator 1 and 11 nucleotides at the 3′ end of the RNA activator 2 together complementarily pair with guide strand of crRNA, and 11 nucleotides at the 3′ end of the RNA activator 1 and 9 nucleotides at the 5′ end of RNA activator 2 are together complementary to the bridge. When the closed loop constructure of crRNA:RNA activator 1:RNA activator 2:bridge quadruplex is formed, it may trigger Cas13a HEPN-nuclease catalytic activity for collateral fluorescent RNA reporter cleavage. Conversely, the base pairing of crRNA with one of the two RNA activators, even after binding to the bridge, cannot activate Cas13a.

To determine whether the crRNA:RNA activator 1:RNA activator 2:bridge quadruplex triggers Cas13a HEPN-nuclease catalytic activity, we tested 21-, 23- and 25-nt RNA activators, which share the identical crRNA binding site and bridge binding site but contain varied length of bumper domain (1, 3, and 5 nt), based on the consideration of steric hindrance of Cas13a:crRNA binary complex to bridge strand (Fig. 2b). Overall, Cas13a can be activated in the presence of two RNA activators and nuclease activation gradually improved with the increasing bumper length of RNA activator. We further determined apparent rates of fluorescent RNA reporter cleavage (normalized to the contiguous complementary target RNA ($T_{20}$)) from each of the resulting time courses. The data revealed the differences in the ability of each set of RNA activators with varied bumper length to activate Cas13a trans-RNA cleavage. The largest defect in activation was caused by 1-nt bumper domain in 21-nt RNA activator, which resulted in an approximate 50-fold reduction in cleavage rate compared to the contiguous complementary target RNA ($T_{20}$) (Fig. 2c). The reported crystal structure revealed that LbuCas13a adopts a bilobed architecture, with the repeat region of the crRNA anchored in the recognition lobe and the guide:RNA activator duplex bound within the channel in the nuclease lobe[19]. We speculate that 1-nt length (0.34 Å) of bumper domain in 21-nt RNA activator is insufficient to escape from the channel of nuclease lobe, resulting in the inability of RNA activators to hybridize with the bridge to form a loop constructure. Furthermore, RNA activators with 5-nt length of bumper domain resulted in similar Cas13a HEPN-nuclease activation relative to contiguous complementary RNA target sequence

($T_{20}$), indicating that partial bumper domain is probably outside the channel and that RNA activators could be fully bind to the bridge, promoting HEPN-nuclease catalytic activity. Based on these results, we speculated that RNA activators with bumper length of 5 or more nucleotides could effectively activate Cas13a. To demonstrate this speculation, we have tested 40-, 60-, and 100-nt RNA activators, containing bumper domains of varying length (20, 40, and 80 nt). Supplementary Fig. 7 shows that there was no significant difference of apparent cleavage rate between different RNA activators with varied bumper length and contiguous complementary target RNA ($T_{20}$) with the variation less than 10%. Then, to obtain the sensitivity of Cas13a in this activation mode, a series of samples containing a set of 25-nt RNA activators and bridge were prepared to initiate the reaction. Figure 2d shows the background with no RNA activators (black line) and typical fluorescence intensity curves in the presence of varying concentrations of RNA activators (1–100 pM). The system was then calibrated versus concentrations of RNA activators, with responses of fluorescence growth rate (Fig. 2e). Using direct fluorescent readout, this system was capable of easily detecting RNA activators as low as 1 pM with the detection limit of 335 fM. Altogether, the realization of two RNA activators directing Cas13a activation with the assistance of an auxiliary strand provides theoretical support and constructive inspiration for the simultaneous and specific detection of two nucleic acids without loss of sensitivity based on the CRISPR-Cas13a system.

## CRISPR-Cas13a Gemini System

Although we have accomplished our noncontiguous target activation paradigm by the assistance of a DNA bridge strand, we were also interested in investigating whether Cas13a could be activated as two target RNAs forming a loop constructure with two Cas13a:crRNA binary complexes. We designed two RNA activators, each of which contained two crRNA binding sites and a bumper domain, and hypothesized that the both ends of two RNA activators may link two distinct Cas13a:crRNA binary complexes to form a fist-to-fist architecture with the crRNA:RNA activator 1:RNA activator 2:crRNA′ quadruplex lying in the channels within the nuclease lobes, termed as CRISPR-Cas13a Gemini System, so that the two Cas13a nucleases could be activated simultaneously and cooperate in RNA cleavage (Fig. 3a).

To determine whether steric hindrance of two Cas13a:crRNA binary complexes affects Cas13a catalytic activity, we tested 21-, 23-, and 25-nt RNA activators, which share the identical crRNA binding sites but contain varied length of bumper domain (1, 3, and 5 nt). The data revealed that Cas13a HEPN-nuclease activation gradually improved with the increasing bumper length of RNA activator and 25-nt RNA activator with 5-nt length of bumper domain resulted in similar Cas13a activation relative to complementary contiguous targets (Target and Target′ corresponding to crRNA and crRNA′, respectively) (Supplementary Fig. 8). The formation of crRNA:RNA activator 1:RNA activator 2:crRNA′ quaternary complex was demonstrated by electrophoretic gel analysis (Supplementary Fig. 9). Our data show that the two RNA activators or crRNAs did not exhibit intermolecular interactions (L3, L6 in Supplementary Fig. 9), crRNA or crRNA′ had varied binding affinities with RNA activators (L7–L12 in Supplementary Fig. 9), and the migration rate of product in lane 13 (L13 in Supplementary Fig. 9) was

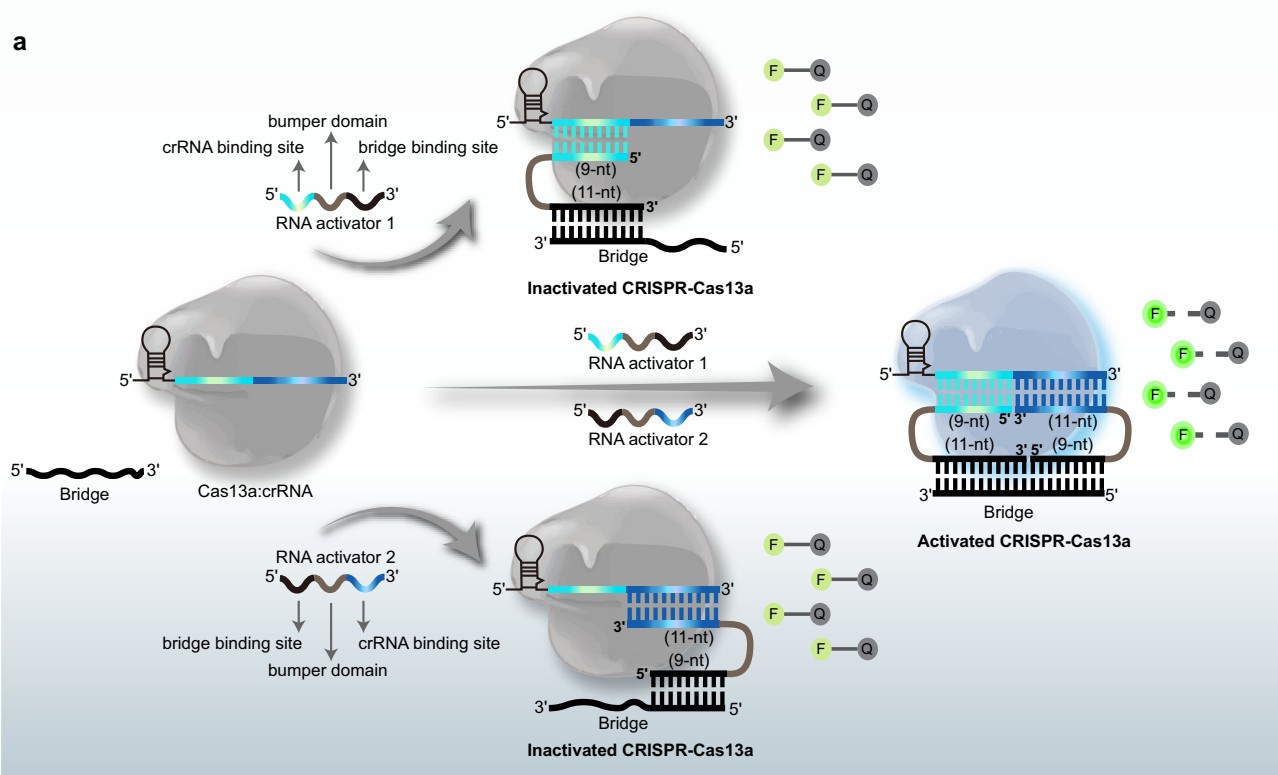

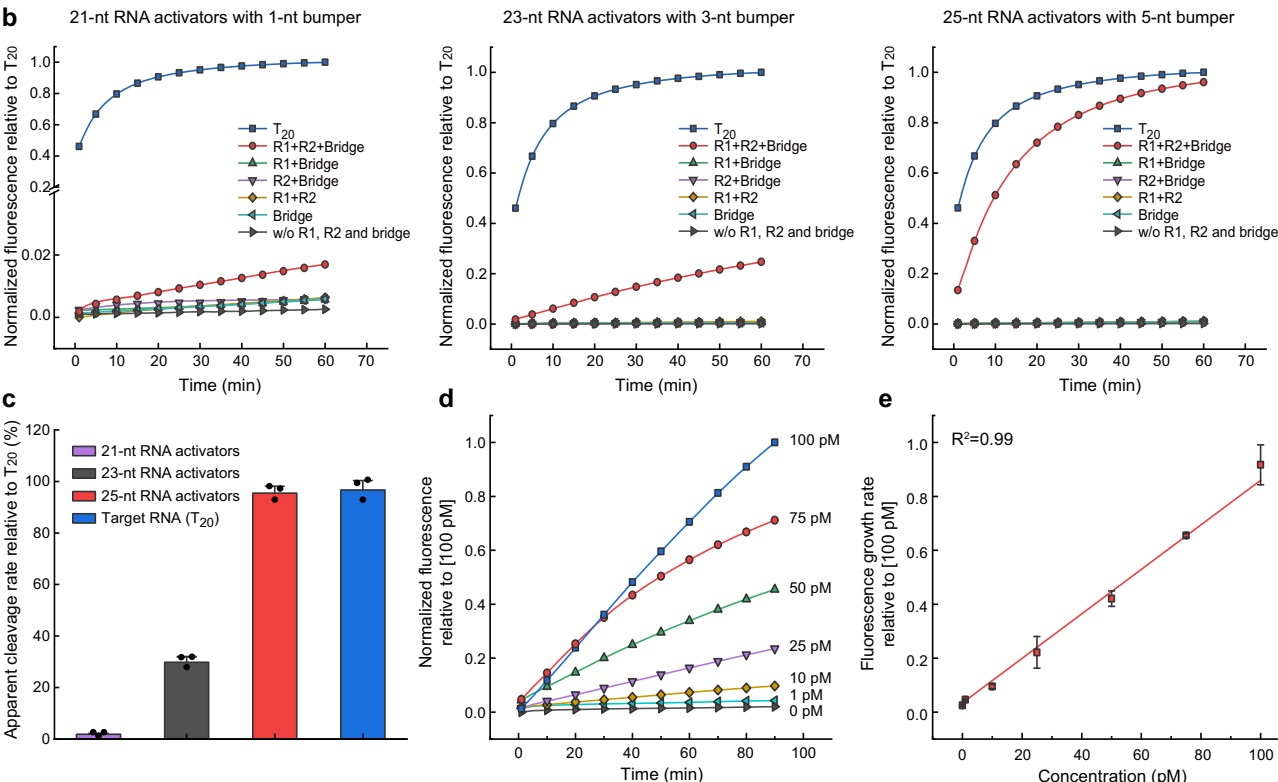

**Fig. 2 | Two RNA activators co-activate Cas13a with the assistance of an auxiliary sequence. a** Schematic of two RNA activators co-activate Cas13a with the assistance of an auxiliary sequence. F-Q represents FAM and BHQ-labeled RNA. **b** Fluorescence analysis of HEPN-nuclease activity of Cas13a by two 21-, 23-, or 25-nt RNA activators. R1 and R2 are abbreviations of RNA activator 1 and RNA activator 2, respectively. **c** Apparent cleavage rate of two 21-, 23-, or 25-nt RNA activators-mediated Cas13a trans-cleavage reporter relative to contiguous target RNA ($T_{20}$). **d** Typical fluorescence intensity curves in the presence of varying concentrations of 25-nt RNA activators (1–100 pM). **e** Derived calibration curve corresponding to the fluorescence changes at different concentrations of 25-nt RNA activators. Data are presented as mean values ± standard deviation from three independent experiments. Source data are provided as a Source data file.

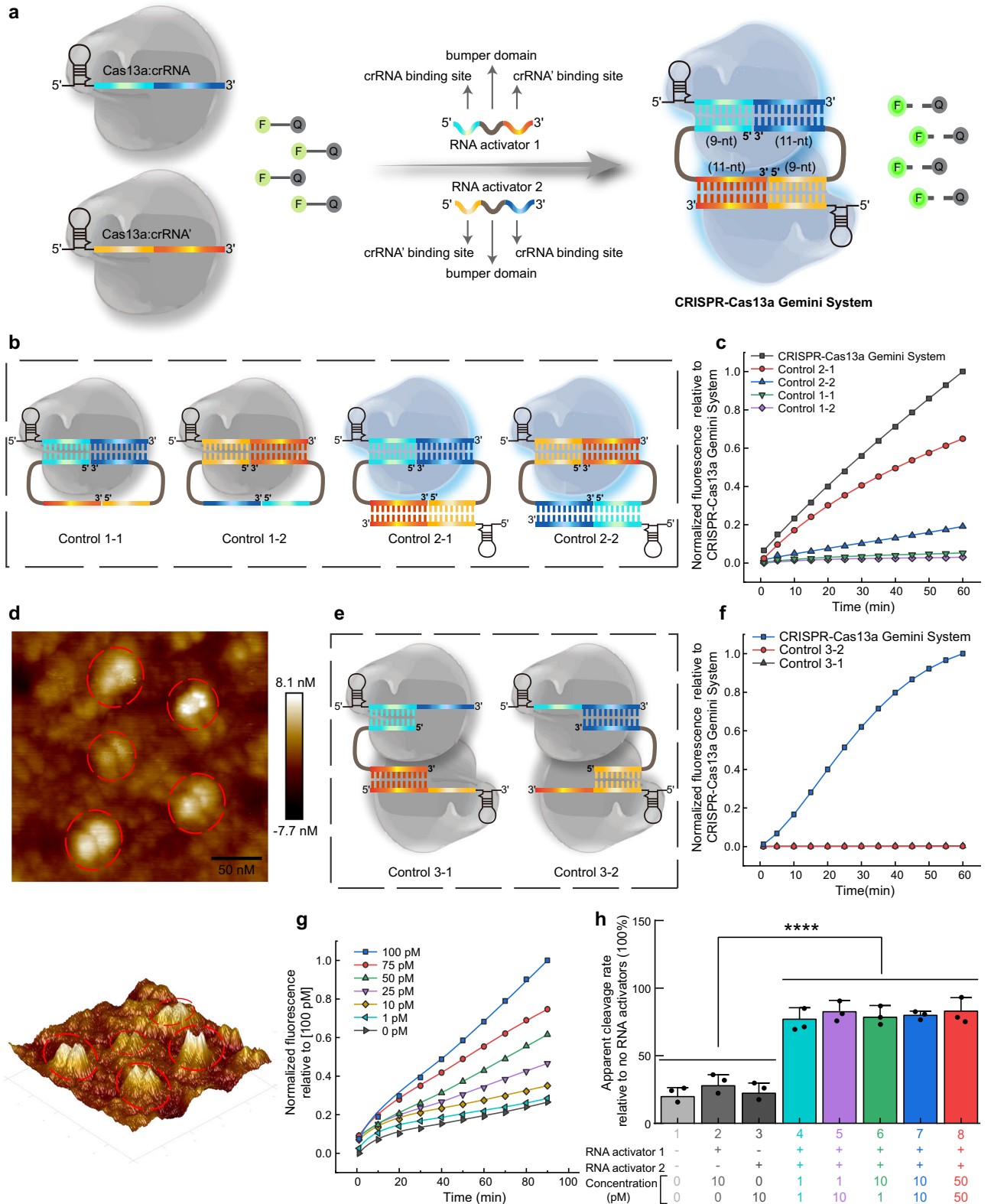

**Fig. 3 | CRISPR-Cas13a Gemini System. a** Schematic of CRISPR-Cas13a Gemini System. **b** The formation of Cas13a:crRNA binary complex either with two RNA activators (Control 1-1, 1-2) or in the presence of two RNA activators and a crRNA bridge (Control 2-1, 2-2). **c** Fluorescence analysis of CRISPR-Cas13a Gemini system and Control 1-1, 1-2, 2-1 and 2-2. CRISPR-Cas13a Gemini system had the highest fluorescence intensity. **d** 2D and 3D AFM images of CRISPR-Cas13a Gemini system. **e** Schematic of one RNA activator controls (Control 3-1, 3-2). **f** Fluorescence analysis one RNA activator controls and CRISPR-Cas13a Gemini system. **g** Typical

fluorescence intensity curves in the presence of varying concentrations of 25-nt RNA activators (1–100 pM). **h** CRISPR-Cas13a Gemini System as a rapid diagnostic platform to distinguish samples containing two RNA targets with varied concentrations and ratios from negative control and samples containing only one RNA target (****$p = 3.3 \times 10^{-15}$, two-tailed $t$ test). Data are presented as mean values ± standard deviation from three independent experiments. Source data are provided as a Source data file.

significantly slower than other lanes, indicating the formation of the longest and most complex product (crRNA:RNA activator 1:RNA activator 2:crRNA' quaternary complex). To verify the synergistic trans-RNA cleavage effect of CRISPR-Cas13a Gemini System, we compared its efficiency of non-target fluorescent RNA cleavage to controls with one set of Cas13a:crRNA binary complex either with two 25-nt RNA activators (Control 1-1, 1-2) or in the presence of two 25-nt RNA activators and a crRNA bridge (Control 2-1, 2-2) (Fig. 3b). These four control experiments resulted in negligible or low increases in fluorescence relative to CRISPR-Cas13a Gemini System (Fig. 3c). Both of Control 2-1 and Control 2-2 could trigger Cas13a HEPN-nuclease catalytic activity, because of the assistance of crRNA' or crRNA as the bridge strand. The possible reasons for the different fluorescence signals between Control 2-1 and Control 2-2 are as follows: The GC contents of complementary bases between RNA activators and crRNA (or crRNA') in Control 2-1 and Control 2-2 are 60% and 50%, respectively (Supplementary Fig. 10). The stronger binding of RNA activators:crRNA triplex in Control 2-1 triggers HEPN1 domain to move closer toward HEPN2 domain, activating the HEPN catalytic site of Cas13a nuclease, which subsequently cleaves more RNA reporters in a non-specific manner, comparing to Control 2-2. Furthermore, several studies have reported that different sequences of guide region in crRNA will affect Cas13a HEPN-nuclease catalytic activity, resulting in differences in fluorescence signals[26,27]. To exclude the possibility of simply crystal packing, we dissolved the crystal of CRISPR-Cas13a Gemini System and checked it by atomic force microscopy (AFM). Many particles contain two Cas13a nucleases positioned next to each other, with a fist-to-fist architecture, demonstrating the formation of Cas13a:crRNA:RNA activator 1:RNA activator 2:crRNA':Cas13a senary complex (Fig. 3d). In contrast, the AFM images of Control 2-1 or Control 2-2 containing one set of Cas13a:crRNA binary complex show single peak morphology (Supplementary Fig. 11). Then, we tested whether one RNA activator containing partial complementary sequence to the guide strand of crRNA and crRNA' can activate Cas13a (Fig. 3e). CRISPR-Cas13a Gemini System catalyzed efficient trans-RNA cleavage only when two RNA activators could base pair with guide sequence in both of crRNA and crRNA' (Fig. 3f). Achieving femtomolar sensitivity (733 fM) (Fig. 3g), CRISPR-Cas13a Gemini System has comparable level of sensitivity as the previous reported CRISPR-Cas13a systems[10,11], whereas, which alone fail to detect two RNAs in a single reaction. Finally, we sought to determine if the CRISPR-Cas13a Gemini System could be adapted as a rapid diagnostic platform like a test strip, which is highly valuable in the current pandemic of respiratory infectious diseases. We found that CRISPR-Cas13a Gemini System could significantly distinguish samples containing two RNA targets with varied concentrations (1, 10, and 50 pM) and ratios (1:10 and 10:1) from negative control and samples containing only one RNA target, but there was no significant difference when the concentrations of two RNA targets were not <1 pM, regardless of their ratio (Fig. 3h).

### CRISPR-Cas13a Gemini System for disease diagnostics

CRISPR-Cas13a Gemini System provides the possibility to detect two RNAs simultaneously in the clinic to increase the accuracy of the single readout. The sequences of the detected RNAs with underlined targeted regions to crRNAs and their corresponding designed crRNAs were presented in Supplementary Table 3. We first employed CRISPR-Cas13a Gemini System to evaluate the abundance of miR-155 and miR-375 in serum samples from 15 healthy subjects, 15 early-stage (stage I–II) and 15 late-stage (stage III–IV) breast cancer (BC) patients with similar age (Fig. 4a and Supplementary Table 4). The levels of miR-155 and miR-375 in patient serum are in the range of picomolar to nanomolar[28,29], which were in the dynamic range of CRISPR-Cas13a Gemini System. The RNA from the serum samples was isolated and purified, using a commonly available RNA purification kit. The violin plots revealed that CRISPR-Cas13a Gemini System

could significantly distinguish stage I–II or stage III–IV BC patients from healthy subjects (Fig. 4b). Receiver operating characteristic (ROC) curves for the CRISPR-Cas13a Gemini System shows area under curve (AUC) values of 0.951 (sensitivity = 86.7%, specificity = 93.3%) and 0.964 (sensitivity = 86.7%, specificity = 100%) in early-stage BC patients (stage I–II) and BC patients (stage I–IV) compared to healthy subjects, respectively, while the results for single RNAs determined by traditional CRISPR-Cas13a system were not comparable (Fig. 4c, Supplementary Fig. 12 and Supplementary Table 5). Next, we utilized CRISPR-Cas13a Gemini System to screen two Epstein-Barr virus (EBV) encoded small RNAs (EBER-1 and EBER-2), from patient samples (75 EBV positives and 52 negatives) (Fig. 4d). The EBERs are expressed at high levels ($10^7$ copies per cell) in cells transformed by EBV[30]. CRISPR-Cas13a Gemini System enabled the identification of 75 positives and 50 negatives, which correspond to the sensitivity, specificity and accuracy of 100%, 96.2%, and 98.4%, respectively (Fig. 4e). Furthermore, this detection achieved a good receiver operating characteristic graph with an AUC of 0.9982 (Fig. 4f and Supplementary Table 6). As a proof-of-concept demonstration of the use in highly efficient and effective dual biomarker detection, CRISPR-Cas13a Gemini System was applied for the EBV detection by a naked-eye lateral flow assay (LFA) (Fig. 4g). We compared the CRISPR-Cas13a Gemini System in fluorescence assay with commercial LFA by testing several clinical samples, both of which successfully distinguished the positive samples from healthy subjects (Fig. 4h and Supplementary Fig. 13). Notably, CRISPR-Cas13a Gemini System in fluorescence assay and LFA generated consistent signals that high fluorescence signals were corresponding to strong visible signals. Taken together, these results demonstrate the capability of CRISPR-Cas13a Gemini System in the highly sensitive and selective disease screening.

### CRISPR-Cas13a Gemini System for dual transgene knockdown

To evaluate the transgene knockdown of two messenger RNAs by CRISPR-Cas13a Gemini System in mammalian cells, we transfected a plasmid coding for EGFP, mCherry and Cas13a together with two EGFP&mCherry-targeting crRNAs, EGFP-targeting crRNAs, mCherry-targeting crRNAs or nontargeting crRNAs into HEK293T cells (Fig. 5a). Using the dual-fluorescence approach, we found that transfection of Cas13a with two EGFP&mCherry-targeting crRNAs co-targeting EGFP and mCherry transcripts resulted in obvious reductions in both EGFP and mCherry fluorescence compared with two crRNAs targeting EGFP or mCherry and nontargeting crRNAs cases (Fig. 5b). Quantitative PCR analysis also revealed significantly decreased levels of both EGFP and mCherry transcripts (Fig. 5c). Overall, our findings reveal another paradigm of Cas13a activation, and provide a theoretical and experimental basis of Cas13a for gene editing of two sites in vivo.

## Discussion

In this work, we present another paradigm of Cas13a activation and established CRISPR-Cas13a Gemini System to implement simultaneous dual nucleic acid detection in a single readout in vitro or effective dual transgene knockdown in vivo. Noncontiguous target RNA activation paradigm of Cas13a was first demonstrated by employing a long RNA sequence, 5' and 3' ends of which together are completely complementary to the guide region of crRNA, forming a loop constructure with only one break site. It is worthwhile to explore whether target RNA with two or more break sites can trigger Cas13a HEPN-nuclease activity in the future. Although non-contiguous RNA binding may slightly affect the conformational change of Cas13a, it does not significantly affect the RNA cleavage activity of Cas13a. On this basis, two target RNAs co-activate Cas13a with the assistance of an auxiliary sequence. Then, two target RNAs form a loop with two Cas13a:crRNA binary complexes to build up

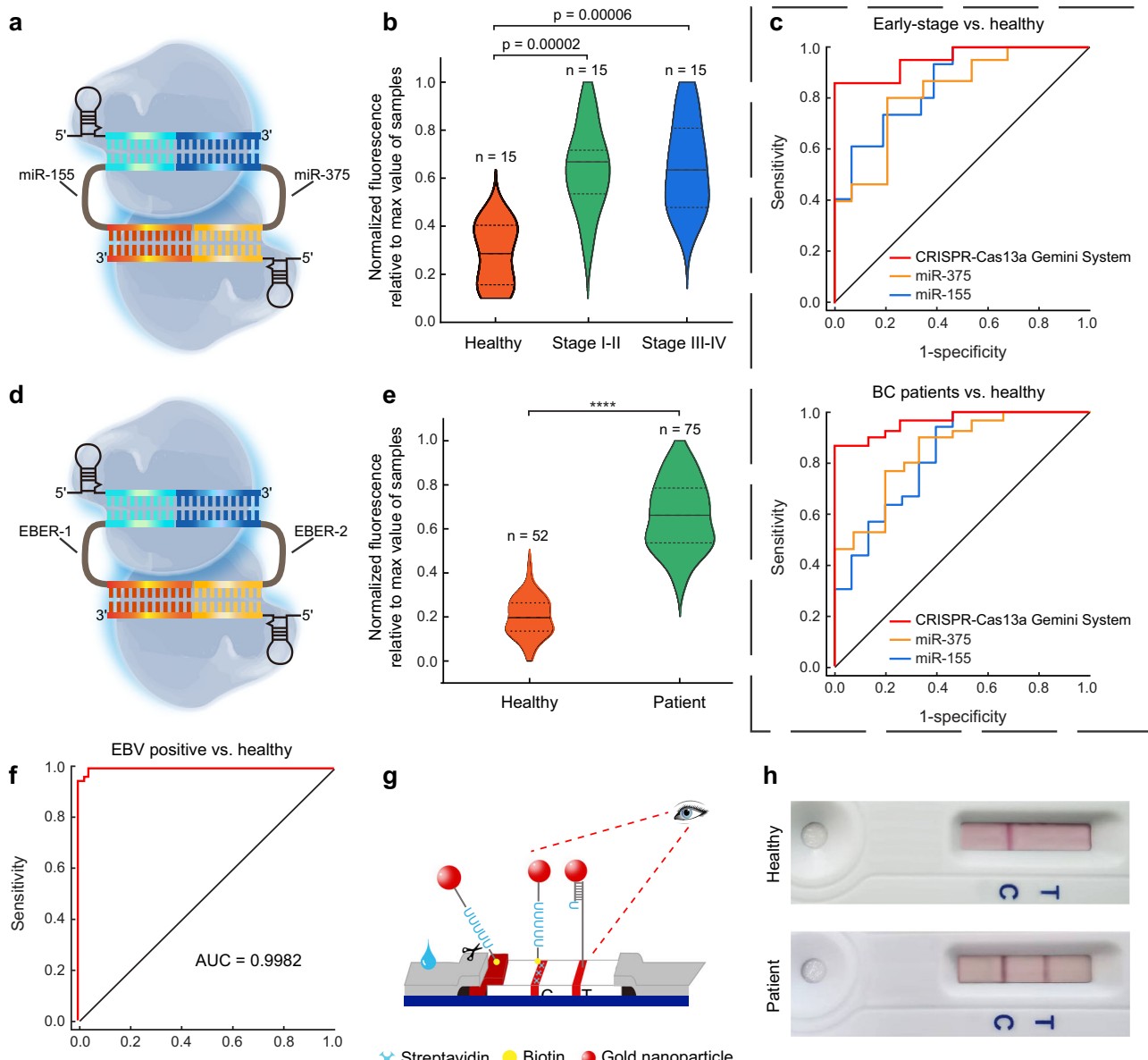

**Fig. 4 | CRISPR-Cas13a Gemini System for disease diagnostics. a** Schematic of CRISPR-Cas13a Gemini System for dual miRNA detection. **b** Statistical quantification of miR-155 and miR-375 expression in the serum of 15 healthy subjects, 15 stage I–II ($p = 0.00002$, two-tailed $t$ test), and 15 stage III–IV ($p = 0.00006$, two-tailed $t$ test) BC patients. **c** ROC curve analysis of early-stage BC patients and BC patients vs. healthy subjects, respectively. Single RNAs were determined by traditional CRISPR-Cas13a system. **d** Schematic of CRISPR-Cas13a Gemini System for dual small RNA detection. **e** Statistical quantification of EBER-1 and EBER-2 expression in the serum of 52 healthy subjects and 75 EBV-positive patients (****$p = 3 \times 10^{-39}$, two-tailed $t$ test). **f** ROC curve analysis of EBV-positive patients vs. healthy subjects. **g** Universal application of CRISPR-Cas13a Gemini System in the detection of EBV on naked-eye lateral flow platform. **h** Detection of healthy and EBV-positive samples by colorimetric lateral flow assay to generate visible signals. Source data are provided as a Source data file.

CRISPR-Cas13a Gemini System that features a fist-to-fist architecture. Our CRISPR-Cas13a Gemini System has comparable level of sensitivity as the previous reported CRISPR-Cas13a systems, whereas, which alone fail to detect two RNAs in a single reaction[10,11]. CRISPR-Cas13a Gemini System allows for simultaneous detection of two microRNAs (miR-155 and miR375) in serum samples of breast cancer patients or two small RNAs (EBER-1 and EBER-2) from Epstein-Barr virus-infected patient samples, significantly distinguishing stage I–II or III–IV breast cancer patients from healthy subjects and precisely identifying Epstein−Barr virus-positive cases, highlighting its clinical potential in early cancer diagnosis and pandemic disease screening. CRISPR-Cas13a Gemini System enables a controllable, highly effective and parallel knockdown of two foreign genes (EGFP and mCherry

mRNAs) in HEK293T cells. The advantages of the CRISPR-Cas13a Gemini System include: simultaneous detection of two RNA targets in the same physical space could increase the accuracy of the single readout and deeply explore the correlation between the two RNA targets. In addition, knockdown of two RNA targets can be achieved at the same time in the same cell. In particular, our method could be quite powerful in the instance that it will produce a positive signal only if both targets are present not one or the other. CRISPR-Cas13a Gemini System represents a next generation CRISPR-Cas13a system, which could potentially be further engineered and reconfigured for detecting and editing three or more nucleic acids by forming a "triangular" structure of Cas13a:crRNA ternary complex or "polygonal" structure of multi-sets of Cas13a:crRNA complexes.

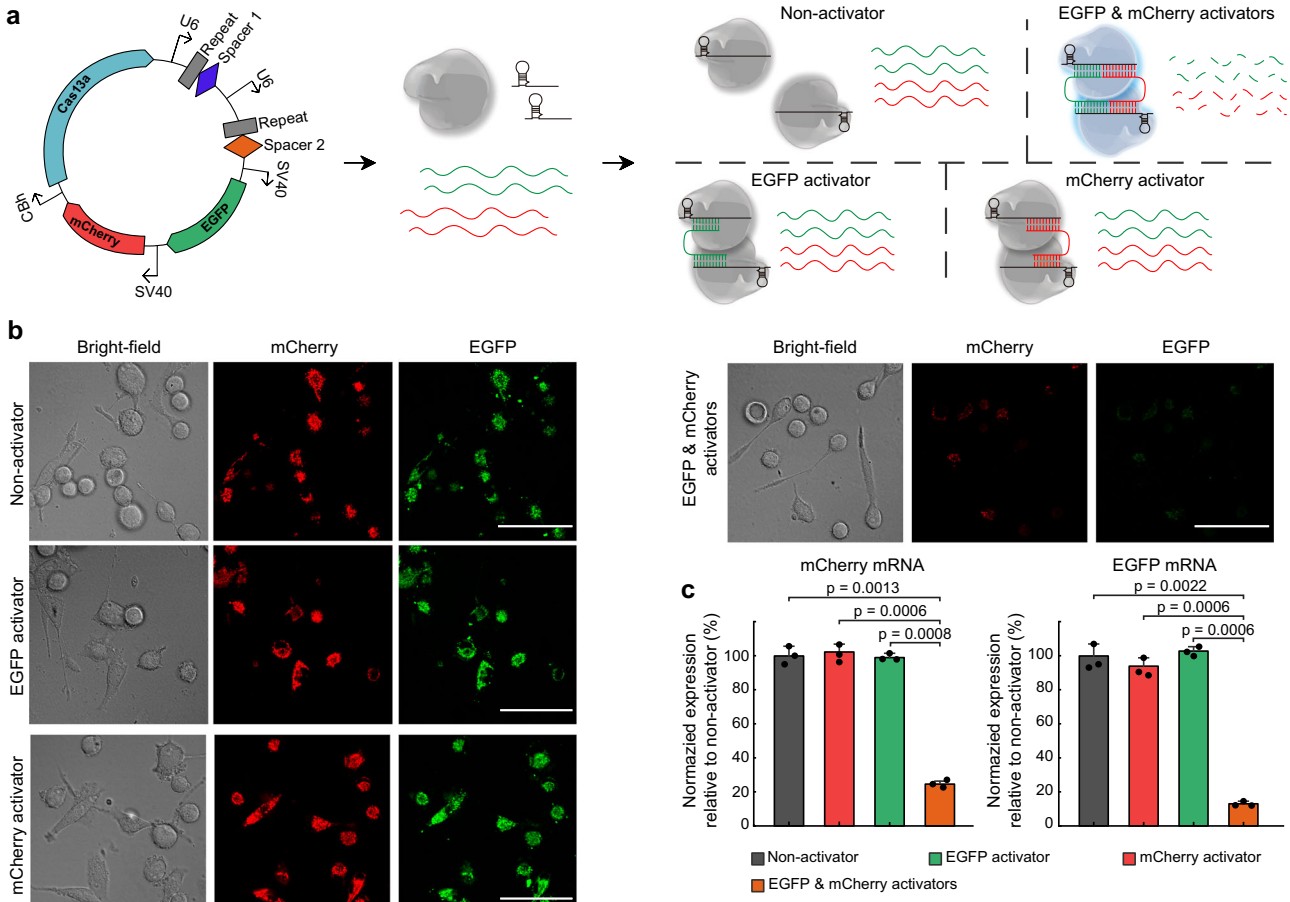

**Fig. 5 | CRISPR-Cas13a Gemini System for two-site gene editing. a** Schematic of the dual-fluorescence reporter system used to evaluate CRISPR-Cas13a Gemini system for parallel gene editing of two messenger RNAs in cells. Qualitative (**b**) and quantitative (**c**) analysis of HEK293T cells with mCherry and EGFP RNA degradation by Cas13a:crRNA with one RNA activator control or CRISPR-Cas13a Gemini System. Scale bar: 50 μm. Data are presented as mean values ± standard deviation from three independent experiments. Two-tailed *t* test was used for statistical analysis. Source data are provided as a Source data file.

# Methods

## Ethical statement

All subjects gave their informed consent for inclusion before they participated in the study. The study was conducted in accordance with the Declaration of Helsinki, and the protocol was approved by the Ethics Committee of South China Normal University (No. SCNU-BIP-2022-015).

## LbuCas13a protein production and purification

Expression vectors deposited with Addgene (Plasmid #83482) were used for expression of LbuCas13a. The codon-optimized Cas13a genomic sequences are N-terminally tagged with a His6-MBP-TEV cleavage site sequence, with expression driven by a T7 promoter. Briefly, expression vectors were transformed into BL21(DE3) cells grown in 2×Terrific broth at 37 °C, induced at mid-log phase (OD600 0.6) with 0.5 mM IPTG, and then transferred to 16 °C for overnight expression of Cas13a. Cells were subsequently harvested, resuspended in lysis buffer (50 mM Tris-HCl pH 7.0, 500 mM NaCl, 5% glycerol, 1 mM TCEP, 0.5 mM PMSF, and Complete-EDTA-free protease inhibitor) and lysed by sonication, and the lysates were clarified by centrifugation at 7000 × *g*. Soluble His-MBP-Cas13a was isolated over metal ion affinity chromatography, and protein-containing eluate was incubated with TEV protease at 4 °C overnight while dialyzing into ion exchange buffer (50 mM Tris-HCl pH 7.0, 250 mM KCl, 5% glycerol, 1 mM TCEP) in order to cleave off the His6-MBP tag. Cleaved protein was loaded onto a HiTrap SP column and eluted over a linear KCl (0.25 to 1 M) gradient. Cation exchange chromatography fractions were pooled and concentrated with 50 kDa cutoff concentrators (Thermo Fisher) and were subsequently flash frozen for storage at −80 °C.

## Fluorescence assay for Cas13a activation by contiguous target RNAs or noncontiguous target RNAs

Cas13a:crRNA complexes were individually preassembled by incubating 1 μL of 500 nM of LbuCas13a protein with 1 μL of 500 nM of crRNA for 5 min at room temperature, followed by 1 μL of rCutSmart™ Buffer (NEB #B6004V), pure water and 1 μL of 20 μM FQ5U reporter. Then, 1 μL of contiguous target RNAs ($T_{20}$, $T_{10}$–$T_{15}$, and $T_{10'}$–$T_{15'}$) or 11 groups of noncontiguous target RNAs were added to the above premix to form the 10 μL reaction system (contiguous target RNAs or noncontiguous target RNAs with 1 nM final concentration). The mix were reacted in a fluorescence plate reader (ABI Thermo Fisher Scientific) at 25 °C for 60 min with fluorescence measurements taken every 1 min ($\lambda_{ex}$: 485 nm; $\lambda_{em}$: 535 nm).

## Gel electrophoresis analysis

In all, 10% polyacrylamide gel electrophoresis (PAGE) gel was used for gel electrophoresis analysis, which were made using 1×TBE and Gel Red as an oligonucleotide dye. LbuCas13a:crRNA:Target RNAs with RNA substrate or crRNA:Target RNA complexes were preassembled in 1×PBS for 60 min at 25 °C and then were mixed with loading buffer. Each lane contains 4 μL of sample-loading complex, run at 120 V for 40 min at room temperature. Gel images were obtained by using the Gel Doc XR+ (Bio-Rad).

## Fluorescence assay for RNA activators co-activate Cas13a with the assistance of an auxiliary sequence

Cas13a:crRNA complexes were pre-assembled individually by incubating amounts of 250 nM LbuCas13a and 250 nM crRNA for 5 min at room temperature. Meanwhile, 21-nt RNA activators (21-nt RNA activator 1 and 21-nt RNA activator 2), 23-nt RNA activators (23-nt RNA activator 1 and 23-nt RNA activator 2) or 25-nt RNA activators (25-nt RNA activator 1 and 25-nt RNA activator 2) and the bridge strand were preassembled together by incubating at the same concentration for 30 min in rCutSmart™ Buffer (NEB #B6004V) at 25 °C, which were then added to Cas13a:crRNA binary complexes for assembly. The final concentrations of 21/23/25-nt RNA activator 1, 21/23/25-nt RNA activator 2, bridge and continuous Target RNA in Fig. 2b, c were 1 nM, respectively. The fluorescence assay was carried out on Quant Studio 3 (Thermo Fisher Scientific) at 25 °C for 120 min with fluorescence measurements taken every 1 min ($\lambda_{ex}$: 485 nm; $\lambda_{em}$: 535 nm).

## Fluorescence assay for CRISPR-Cas13a Gemini System

In CRISPR-Cas13a Gemini System, Cas13a:crRNA complexes and Cas13a:crRNA' complexes were individually preassembled by incubating 150 nM of LbuCas13a with 150 nM of crRNA or crRNA' for 5 min at 25 °C, then mixing the two complexes together and followed by the addition of 25-nt RNA activator 1 and 25-nt RNA activator 2 to the complexes and incubation for 30 min in rCutSmart™ Buffer (NEB #B6004V) at 25 °C to complete the assembly of CRISPR-Cas13a Gemini System. Then add 20 µM FQ5U into this system and replenish 10 µL of final system with RNase-free water. 21-nt RNA activators and 23-nt RNA activators were also tested using the above experimental method. In Control 2-1 (or Control 2-2), Cas13a:crRNA complex (or Cas13a:crRNA' complex) was preassembled by incubating 150 nM of LbuCas13a with 150 nM of crRNA (or crRNA'), followed by the addition of RNA activators and crRNA' (or crRNA). The reactions were incubated in Quant Studio 3 (Thermo Fisher Scientific) at 25 °C for 120 min with fluorescence measurements taken every 1 min ($\lambda_{ex}$: 485 nm; $\lambda_{em}$: 535 nm).

## Atomic force microscopy characterization

Cas13a:crRNA complexes and Cas13a:crRNA' complexes were individually preassembled by incubating 1 µM of LbuCas13a with 1 µM of crRNA or crRNA' for 5 min at room temperature, then mixing the two complexes together and followed by adding 1 µM of 25-nt RNA activator 1 and 25-nt RNA activator 2 to the complexes and incubated for 60 min at 25 °C in 1× PBS buffer. The samples were then sent to eceshi company for AFM photography. AFM images were taken on Bruker Dimension Icon AFM, and the images were processed and analyzed using NanoScope Analysis 1.9.

## Plasmid construction, cell culture, transfection, and fluorescence analysis

In the experiments referred to in Fig. 5, the plasmid used was constructed as follows: CBh-Cas13a-U6-crRNA1-U6-crRNA2-SV40-EGFP-SV40-mCherry. HEK293T cells (Chinese Academy of Sciences Cell Bank) well cultured with DMEM (Thermo Fisher Scientific) supplemented with 10% fetal bovine serum (Thermo Fisher Scientific), 1% penicillin–streptomycin (Thermo Fisher Scientific) and 0.1 mM non-essential amino acids (Thermo Fisher Scientific) in an incubator at 37 °C with 5% $CO_2$. In all, $3 \times 10^4$ HEK293T cells per well were plated in the 48-well plate in 0.25 mL of complete growth medium. After 12 h, 0.5 µg plasmids were transfected into cells with 1.5 µg poly-ethylenimine (PEI) (DNA/PEI ratio of 1:3). Forty-eight hours after transfection, EGFP and mCherry fluorescence were analyzed by confocal laser scanning microscopy (AX, Nikon, Japan).

## Serum sample preparation and RNA isolation

All samples were collected with the informed consent of the patients, and the study was conducted with the approval of the Internal Review Boards of the designated hospital. Inclusion criteria for patients was at least 18 years of age. Serum samples were obtained from patients and healthy donors as follows: Venous blood samples were collected by authorized hospital staff. Whole blood samples were then clotted by incubation at room temperature for 30 min, centrifuged at 4 °C and 1500 × g for 10 min to remove clots, and the supernatant obtained is serum (serum samples were obtained from The First Affiliated Hospital, Sun Yat-sen University). RNA was isolated from serum samples using the RNeasy Kit (QIAGEN) following the manufacturer's protocol. All samples were randomly selected from a larger cohort and analyzed. Unblinding of clinical parameters and corresponding experimental data was performed only after completion of all experiments.

## Quantitative PCR measurement of RNA

qRT-PCR reactions were performed using the RNA qRT-PCR TB Green kit (Takara). Perform reverse transcription in a solution containing 3.75 µL RNA sample, 1.25 µL enzyme mix (poly(A) polymerase, MMLV Reverse Transcriptase) and 5 µL 2× reaction buffer. RT reactions were performed at 37 °C for 1 h, then heated to 85 °C for 5 min to inactivate the enzyme. The qPCR thermocycling program was: 95 °C for 10 s followed by 40 amplification reaction cycles at 95 °C for 5 s, 60 °C for 20 s, with a dissociation step. Fluorescence signals were recorded by Quant Studio 3 (Thermo Fisher Scientific).

## Statistics and reproducibility

Statistical tests were performed using GraphPad Prism version 8.0, IBM SPSS Statistics version 19.0 and Student's $t$ test. The statistical test used for the data shown in each figure is noted in the corresponding figure legend, and significant statistical differences are noted as exact $P$ values. All values are reported as mean values ± standard deviation from three independent experiments. No data were excluded from the analyses.

## Reporting summary

Further information on research design is available in the Nature Portfolio Reporting Summary linked to this article.

## Data availability

All the data generated in this study are available with this manuscript and its Supplementary Information. Source data have been deposited in Figshare[31]; https://doi.org/10.6084/m9.figshare.25097423. Source data are provided with this paper.

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

## Acknowledgements
This work was supported financially by the National Key Research and Development Program of China (2023YFC2417200, J.H.), Guangdong Provincial Pearl River Talents Program (2019QN01Y725, J.H.), Science and Technology Program of Guangzhou (202206010108, J.H.), Natural Science Foundation of Guangdong Province (2022A1515010370, J.H.; 2022A1515011721, Y.S.), Recruitment Program of Global Experts (J.H.), The Major Program of Ningbo Science and Technology Innovation 2025 (2020Z093, J.H.). The schematic diagram of the Cas13a protein shown in Figs. 1a, 2a, 3a, b, e, 4a, d, 5a and Supplementary Figs. 3a, 6a, c, e, and 7a was reproduced with permission from Fig. 1 of ref. 18. Copyright 2019 American Chemical Society.

## Author contributions
J.H. and Y.S. conceived the concept; H.Z. and Y.S. designed the experiments; H.Z., Y.S., T.Z., S.Z., Y.Z., F.Q., M.L. and W.X. performed the experiments; H.Z. analyzed data and wrote the paper; J.H., Y.S. and D.Z. commented and revised the paper.

## Competing interests
The authors declare no competing interests.
