## [Peer Review File · Nature Communications]

REVIEWER COMMENTS

Reviewer #1 (Remarks to the Author):

Zhao et al. describes an innovative gRNA design against noncontiguous targets. They systematically identified design rules to achieve Cas13a activation by two different RNAs, with sensitivity comparable to the conventional Cas13a-mediated detection method that is only capable of detecting one target. This paper also shows the potential of CRISPR-Cas13a Gemini System in various applications, such as rapid and sensitive detection of two miRNAs for breast cancer diagnosis and two small RNAs for Epstein-Barr virus diagnosis. Overall, this is an innovative design that could have some interesting applications in diagnostics and gene regulation. I think this is suitable for publication in Nature Communications after addressing the following comments.

Major:

1. The spacer sequence is restricted to the 5' and 3' end of any RNA targets, which might not always be the most specific and effective for Cas13 activation and specificity. I wonder about the generalizability of such a detection method.
2. An important control I think is missing from Figure One is an RNA target that possesses a single crRNA targeting region. In contrast, the other targeting region is filled or substituted by a non-targeted sequence.
3. The last section is not gene editing. The authors are cleaving EGFP and mCherry RNA and achieved effective dual transgene knockdown.
4. The bumper length was studied, but what about the loop region in strategy 2? Are the sizes comparable? And how comparable is it to actual therapeutically relevant targets like those in Figure 4

Minor:

Supplementary figure 1a and b are not labeled and a is not mentioned in the text.

Figure resolution is a bit low.

Supplementary figure 2a: target20 is not split at 9.

The statement in line 120 requires some data to back it up. Is binding affinity different between 10 and 8/9 nucleotides? And the conclusion in line 120 is a little confusing.

In lines 130-133, the authors might be able to learn about this hypothesis more if they include a control with filler sequences, as suggested in point 2 of the major questions

Line 128-130: 5' and 3' should be reversed.

Line 144-145: while the authors hypothesis (crRNA sequence is different) is also valid, but there are more variables, and a sequence filler single spacer control can also help. And is there a reason why the crRNA direct repeat used here is different from what's published previously?

Figure 2a can be illustrated with 9 and 11 nucleotides so that Line 167-171 may be easier to understand.

Figure 2 lacks a control where only R1 + R2 are added

Line 182: the statement about Figure 2b saying nuclease activity increases gradually from 21 to 25nt bumper region is not supported by the data.

Figure 2b doesn't quite agree with 2c. What is data in Figure 2b normalized to (same question for all other normalized data except for 2c)?

Line 188: Figure 2c does not show 50% reduction in the 21nt group compared to the full length.

Line 253: ctrl 2-1 is high. Do the authors have any hypothesis? Is it relevant in a potential therapeutic application?

Line 270: what would happen if the 2 RNA activators are not 1 to 1 ratio?

Figure 4C: what are single RNA controls exactly? Are they single crRNA controls? why are these generating strong signals compared to what observed in figure 3 with a testing case? What are some factors that can contribute to this difference between testing cases and real-life application settings?

Reviewer #3 (Remarks to the Author):

Zhao and colleagues present a method named CRISPR-Cas13a Gemini, a method to activate Cas13 cleavage with two different target RNAs that can be harnessed for diagnostics and degrading RNAs in cell culture. They identify the proper design of crRNAs that creates a loop-like structure that brings together two Cas13a-crRNA complexes and activates cleavage in the presence of two separate target RNAs. They apply the Gemini method for the detection of micro RNAs, Epstein Barr RNAs and cleavage of GFP and RFP simultaneously in transfected cells.

I applaud the authors for the following strengths of their manuscript:

1. The schematics at the start of each figure that nicely depict how the developed is intended to active Cas13 cleavage.
2. Several applications of Gemini to demonstrate its versatility.
3. Inclusion of Fig 1f to demonstrate the potential background of partial crRNA:target binding.

Despite these strengths I have the following concerns:

Major

1. Although Gemini represents a new strategy to activate Cas13 against multiple RNAs, it is unclear from the data and the discussion why this approach is more advantageous than providing two crRNAs that fully match two targets compared to splitting the crRNA complementarity across two RNA targets and thereby requiring a bumper region. In Figure 1, T20 is much more efficient at Cas13 (i.e. faster fluorescence accumulation) than both strategies, so the authors should directly compare in Figure 3 through 5 how Gemini compares to providing two RNA targets one that matches crRNA and one that matches crRNA'. This comparison is critical to understanding the relevance of this approach.
2. Given the data in Figure 1, the authors move to a modification of Strategy II in Figure 2 and use a bridge RNA with a region of the activator that binds the bridge as opposed to using a loop, but a key control is missing in Figure 2b (R1+R2 w/o bridge) as it will directly demonstrate that the extra sequence on the activators doesn't drastically change the efficiency and demonstrate the essential addition of the bridge to activate Cas13 efficiently.
3. It is unclear what is the difference between Gemini and Control 2-1 and Control 2-2. Does it relate to pre-incubation of Cas13 with the crRNAs or activators? Is the concentration of Cas13 different so that there are not as many dual Cas13 complexes? AFM images showing lack of dual peaks would support this difference. Also given the sequence similarities between Control 2-1 and 2-2 the authors should comment on why the background fluorescence is so different.

4. Many Cas13 detection methods require pre-amplification e.g. isothermal methods or PCR. The methods do not indicate this was performed for the data presented in Figure 4, but it is unclear what the levels of the microRNAs and EBERs are in patient samples, and as a result the authors either need to measure this by qPCR or assess the limit of detection of their method. Furthermore, only showing one lateral flow for each sample group is not convincing.

5. Figure 4 and 5 are missing key details on the design of the activators. Because of the bridge and crRNA binding requirements do the complementary regions need to be on the end of the RNAs? These design details should be described in the text as they are key to others being able to apply this method to other targets.

Minor

1. Throughout the paper the normalization of the data should be explained in the figure legends.

2. Figure 1D: Legend is more easily interpreted if ordered sequentially by strategy I followed by strategy II and adding SI and SII labels because the subscripts are subtle.

3. Figure 2b: Legend should be R1, R2, and bridge (not or)

4. Line 370-371 claim: reference?

5. The authors should be more cautious throughout when proposing that their method detects two RNA targets (in the case of miRNAs and EBERs) because the way it is phrased it could be misinterpreted to be a multiplexed diagnostic that differentially detects two targets, but in fact it is that Cas13 is activated by two targets increasing the sensitivity of the single readout.

We appreciate all reviewers for their valuable comments and suggestions. Our co-authors have conducted a series of experiments to address the raised concerns in the past two months. Our response to review comments together with new experimental results is given in this letter and the revised manuscript (marked in red). In the following, we present our response (marked in blue) to each review comment (marked in yellow) in detail.

REVIEWER COMMENTS

Reviewer #1 (Remarks to the Author):

Zhao et al. describes an innovative gRNA design against noncontiguous targets. They systematically identified design rules to achieve Cas13a activation by two different RNAs, with sensitivity comparable to the conventional Cas13a-mediated detection method that is only capable of detecting one target. This paper also shows the potential of CRISPR-Cas13a Gemini System in various applications, such as rapid and sensitive detection of two miRNAs for breast cancer diagnosis and two small RNAs for Epstein-Barr virus diagnosis. Overall, this is an innovative design that could have some interesting applications in diagnostics and gene regulation. I think this is suitable for publication in Nature Communications after addressing the following comments.

Major:

1. The spacer sequence is restricted to the 5' and 3' end of any RNA targets, which might not always be the most specific and effective for Cas13 activation and specificity. I wonder about the generalizability of such a detection method.

Answer: We thank the reviewer for this concern. Besides the strategy I and II in **Fig. 1a**, three additional models have been designed to test the generalizability of our method (**Supplementary Fig. 6**). The length of the RNA sequence used in strategy II was 60-nt. Limited to the current nucleic acid synthesis technology, the purity of synthesized nucleic acid sequence over 100-nt will decrease, therefore, the full length of the

synthesized long RNA sequence used in these three models was set at no more than 100-nt.

In model 1A, a short RNA sequence and the target region near 5' end (10, 20, or 40-nt distance from 5' end) of another short RNA sequence together are complementary to the guide region of crRNA. In model 1B, the 3' end and the target region near 5' end (10, 20, or 40-nt distance from 5' end) of a long RNA sequence together are completely complementary to the guide region of crRNA (**Supplementary Fig. 6a**).

In model 2A, a short RNA sequence and the target region near 3' end (10, 20, or 40-nt distance from 3' end) of another short RNA sequence together are complementary to the guide region of crRNA. In model 2B, the 5' end and the target region near 3' end (10, 20, or 40-nt distance from 3' end) of a long RNA sequence together are completely complementary to the guide region of crRNA (**Supplementary Fig. 6c**).

In model 3A, the target region near 5' end (10, 20, or 40-nt distance from 5' end) of a short RNA sequence, and the target region near 3' end (10, 20, or 40-nt distance from 3' end) of another short RNA sequence together are complementary to the guide region of crRNA. In model 3B, the target region near 5' end (10, 20, or 40-nt distance from 5' end), and the target region near 3' end (10, 20, or 40-nt distance from 3' end) of a long RNA sequence together are completely complementary to the guide region of crRNA (**Supplementary Fig. 6e**).

Comparison experiments between short RNA sequences (model 1A, 2A and 3A) and long RNA sequences (model 1B, 2B and 3B) with different locations of target region were performed using G₉₋₁₁ (**Supplementary Fig. 6b, d, f**). The results show that short RNA sequences in these three models exhibited no Cas13a trans-RNA cleavage regardless of the non-target extension sequences at the 5' and/or 3' end, but Cas13a HEPN-nuclease activity could be triggered when forming a loop constructure with the long RNA sequence regardless of whether the target region is at the end or other locations of the sequence. These additional models demonstrate the spacer sequence of crRNA is not restricted to the 5' and 3' end of any RNA targets, indicating the broad generalizability of our method. **Supplementary Fig. 6** has been added in the supplementary information, and the associated description has been added on **page 8** in the revised manuscript.

Supplementary Fig. 6 Noncontiguous target RNA activation paradigm of Cas13a. **a**, Schematic of model 1 for activating Cas13a. **b**, Apparent cleavage rate of model 1 for Cas13a trans-cleavage relative to T_{20} . **c**, Schematic of model 2 for activating Cas13a. **d**, Apparent cleavage rate of model 2 for Cas13a trans-cleavage relative to T_{20} . **e**, Schematic of model 3 for activating Cas13a. **f**, Apparent cleavage rate of model 3 for Cas13a trans-cleavage relative to T_{20} .

2. An important control I think is missing from Figure One is an RNA target that possesses a single crRNA targeting region. In contrast, the other targeting region is filled or substituted by a non-targeted sequence.

Answer: We thank the reviewer for this valuable suggestion, which will make our noncontiguous target RNA activation paradigm of CRISPR-Cas13a more convincing. We established two control groups that a long RNA sequence contains varied complementary sequence (5 to 15-nt) to the guide strand of crRNA at its 5' end and non-targeted sequence at its 3' end ($T_{n+non'}$), and vice versa ($T_{non+n'}$) (**Supplementary Fig. 3a, Supplementary Table 2**). Since these two control groups possess only single crRNA targeting region, we compared their efficiencies of non-target fluorescent RNA cleavage to strategy I (**Supplementary Fig. 3b**). Comparative analysis reveals that $T_{non+n'}$ was unable to activate Cas13a for collateral RNA cleavage. This result is consistent with the conclusion of **Fig. 1f** that target RNA shortened by 5-nt or more from its 3' end ($T_{15'}$ to $T_{10'}$) was unable to activate Cas13a for collateral RNA cleavage. $T_{n+non'}$ with no more than 12-nt complementary sequence from its 5' end failed to activate HEPN-nuclease activity of Cas13a, while $T_{n+non'}$ with 13 to 15-nt complementary sequence from its 5' end exhibits abilities for activating Cas13a, but with lower cleavage rate compared to strategy I. In addition, the cleavage rate of $T_{15+non'}$ is close to $T_{15+5'}$ of strategy II, which demonstrate our hypothesis that a long extension at the 3' end of crRNA- $T_{15+5'}$ duplex might weaken the Cas13a nuclease activation. We have added **Supplementary Fig. 3** and **Table 2** in the supplementary information, and associated description on **page 7** in the revised manuscript.

Supplementary Fig. 3 RNA activator controls that possess a single crRNA targeting region. a, Schematic of RNA activator controls ($T_{n+non'}$, $T_{non+n'}$). **b,** Apparent cleavage rate of RNA activator controls ($T_{n+non'}$, $T_{non+n'}$) for Cas13a trans-cleavage relative to T_{20} .

3. The last section is not gene editing. The authors are cleaving EGFP and mCherry RNA and achieved effective dual transgene knockdown.

Answer: We agree with this comment and have changed ‘gene editing’ to ‘transgene knockdown’ in the whole revised manuscript.

4. The bumper length was studied, but what about the loop region in strategy 2? Are the sizes comparable? And how comparable is it to actual therapeutically relevant targets like those in Figure 4

Answer: We thank the reviewer for this concern. The length of the long RNA sequence used in strategy 2 is 60-nt. Except for the complementary sequence (20-nt) to the guide region of crRNA, the length of the loop region in the long RNA sequence is 40-nt. By subtracting the number of nucleotides (20-nt) corresponding to the bridge strand used in Fig. 2, the bumper length on each side in the long RNA sequence of strategy 2 is 10-nt.

When we first established our system, in order to verify its feasibility, we used a relatively long bumper length in strategy 2, which was 10-nt. After the CRISPR-Cas13a Gemini System was established, we applied it to the actual dual RNA detection (trans-RNA cleavage) or dual transgene knockdown (cis-target cleavage). Since most miRNAs are approximately 21 to 25 nucleotides in length, as shown in **Fig. 2** and **Supplementary Fig. 8**, we investigated the bumper length in RNA activator ranging from 1 to 5-nt (the total number of complementary bases between RNA activator and crRNAs of CRISPR-Cas13a Gemini System is 20).

The actual miRNA targets tested in **Fig. 4** were miR-155 (22-nt) and miR-375 (24-nt), the bumper lengths of miR-155 and miR-375 are 2-nt and 4-nt, respectively. They are comparable to the RNA activators studied in **Fig. 2** and **Fig. 3**, which have been demonstrated for activating CRISPR-Cas13a Gemini System. The actual small RNAs tested in **Fig. 4** were EBER-1 (167-nt) and EBER-2 (173-nt), the bumper lengths of EBER-1 and EBER-2 are 143-nt and 149-nt, respectively. The sequences of these miRNAs and small RNAs with underlined targeted regions to crRNAs have been added in **Supplementary Table 3**. The results in **Fig. 2c** and **Supplementary Fig. 8** revealed that Cas13a HEPN-nuclease activation gradually improved with the increasing bumper length of RNA activator from 1 to 5-nt, in addition, 25-nt RNA activator with 5-nt length of bumper domain and 60-nt long RNA sequence with 10-nt bumper domain in strategy 2 (T_{9+11} , in Fig. 1e) resulted in similar Cas13a activation relative to complementary contiguous targets. Based on these results, we speculated that RNA activators with bumper length of 5 or more nucleotides could effectively activate Cas13a. To demonstrate this speculation, we have tested 40-, 60, and 100-nt RNA activators, containing bumper domains of varying length (20, 40 and 80-nt), somehow comparable to small RNAs in size. **Supplementary Fig. 7** shows that there was no significant difference of apparent cleavage rate between different RNA activators with varied bumper length and contiguous complementary target RNA (T_{20}) with the variation less than 10%. This data demonstrates the capability of our approach to detect long RNA targets. We have added **Supplementary Fig. 7** and **Table 3** in the supplementary information, and associated description on **page 12** in the revised manuscript.

Supplementary Fig. 7 Two RNA activators co-activate Cas13a with the assistance of an auxiliary sequence. a, Schematic of two RNA activators with varied bumper length (20, 40, 80-nt) co-activate Cas13a with the assistance of an auxiliary sequence. **b**, Fluorescence analysis of HEPN-nuclease activity of Cas13a by two 40-, 60- or 100-nt RNA activators. **c**, Apparent cleavage rate of two 40-, 60- or 100-nt RNA activators-mediated Cas13a trans-cleavage relative to contiguous target RNA (T_{20}).

Minor:

Supplementary figure 1a and b are not labeled and a is not mentioned in the text.

Answer: Thanks for this reminder. **Supplementary Fig. 1a** and **b** have been labeled in the **Supplementary Fig. 1**, and also mentioned on **page 5** in the revised manuscript.

Figure resolution is a bit low.

Answer: We thank the reviewer for this concern. The quality of all figures has been improved and the figure resolution has been increased to 500 dpi in the revised manuscript.

Supplementary figure 2a: target20 is not split at 9.

Answer: We thank the reviewer for this suggestion. We have made this correction in the revised supplementary information.

The statement in line 120 requires some data to back it up. Is binding affinity different between 10 and 8/9 nucleotides? And the conclusion in line 120 is a little confusing.

Answer: We thank the reviewer for this suggestion. The simulation and calculation of NUPACK show that the ΔG of 8, 9 and 10 nucleotides bound to crRNA at room temperature are -25.74, -26.41 and -29.23 kcal/mol, respectively. These data suggest that 10 nucleotides was bound more tightly by Cas13a:crRNA than 8 or 9 nucleotides. The conclusion in line 120 has been rewritten as ‘Cas13a target RNA binding affinity and nuclease activation are decoupled and differentially affected by break sites between the target RNA and the guide region of crRNA’. We have added these data and descriptions on **page 6** in the revised manuscript.

In lines 130-133, the authors might be able to learn about this hypothesis more if they include a control with filler sequences, as suggested in point 2 of the major questions

Answer: We thank the reviewer for this valuable suggestion. A control with filler sequences has been included in the revised manuscript, as suggested in point 2 of the major questions.

Line 128-130: 5' and 3' should be reversed.

Answer: After careful examination, as shown in **Fig. L1**, the 15 nucleotides bound to the guide region of crRNA are at the 5' end of T_{15+5'}, and the other 5 nucleotides with weak binding affinities to crRNA are at the 3' end of T_{15+5'}. We have rewritten this sentence for clarity without reversing the 5' and 3' on **page 6** in the revised manuscript.

Fig. L1 The formation of Cas13a:crRNA binary complex with T_{15+5'}.

Line 144-145: while the authors hypothesis (crRNA sequence is different) is also valid, but there are more variables, and a sequence filler single spacer control can also help. And is there a reason why the crRNA direct repeat used here is different from what's published previously?

Answer: We thank the reviewer for the concern on this issue. Both crRNA with a missing base at the 3' end of its repeat region (crRNA[#]) and crRNA with adenine base at the 3' end of its repeat region (crRNA-A) have been reported previously¹⁻⁵. And target RNA shorter than 16-nt in length failed to trigger Cas13a HEPN-nuclease activity in the case of crRNA-A¹. But for crRNA[#], there has been no report on the shortest length requirement of target RNA for Cas13a activation. In the original manuscript, we found that T₁₄ and T₁₅ could activate Cas13a when crRNA[#] was used. According to the reviewer's suggestion, to investigate the influence of the base at the 3' end of the crRNA repeat region on Cas13a activity, we compared 4 crRNAs, including crRNAs with 4 different bases at the 3' end of the repeat region (crRNA-A, crRNA-U, crRNA-C and crRNA-G), with crRNA[#]. The data show that T₁₄ and T₁₅ could effectively activate Cas13a after binding with crRNA[#]. By contrast, crRNA-C group had a weak signal, and neither T₁₄ nor T₁₅ could activate Cas13a in the presence of crRNA-A, crRNA-U or crRNA-G. These results update the shortest length requirement of target RNA for activating the CRISPR-Cas13a system. We have added **Supplementary Fig. 5** in the supplementary information, and associated description on **page 8** in the revised manuscript.

Supplementary Fig. 5 Difference in crRNA sequence. a, Sequences of the designed crRNAs (crRNA#,

crRNA-A, crRNA-U, crRNA-C, crRNA-G) with the same guide region but different in repeat region. **b**,

Fluorescence analysis of Cas13a activation with designed crRNAs by T₁₄ and T₁₅, respectively.

1. East-Seletsky, A. et al. Two distinct RNase activities of CRISPR-C2c2 enable guide-RNA processing and RNA detection. *Nature* 538, 270-273 (2016).
2. East-Seletsky, A., O'Connell, M.R., Burstein, D., Knott, G.J. & Doudna, J.A. RNA Targeting by Functionally Orthogonal Type VI-A CRISPR-Cas Enzymes. *Molecular Cell* 66, 373-383 (2017).
3. Tambe, A., East-Seletsky, A., Knott, G.J., Doudna, J.A. & O'Connell, M.R. RNA Binding and HEPN-Nuclease Activation Are Decoupled in CRISPR-Cas13a. *Cell Rep* 24, 1025-1036 (2018).
4. Fozouni, P. et al. Amplification-free detection of SARS-CoV-2 with CRISPR-Cas13a and mobile phone microscopy. *Cell* 184, 323-333.e329 (2021).
5. Liu, T.Y. et al. Accelerated RNA detection using tandem CRISPR nucleases. *Nature Chemical Biology* 17, 982-988 (2021).

Figure 2a can be illustrated with 9 and 11 nucleotides so that Line 167-171 may be easier to understand.

Answer: We thank the reviewer for this constructive suggestion. **Fig. 2a** has been illustrated with 9 and 11 nucleotides in the revised manuscript.

Figure 2 lacks a control where only R1 + R2 are added

Answer: According to the reviewer's suggestion, a control where only R1 + R2 has been added in **Fig. 2b**. The data reveals that only two RNA activators (R1+R2) were not able to trigger the Cas13a HEPN-nuclease activity. We have added the updated **Fig. 2b** in the revised manuscript.

Line 182: the statement about Figure 2b saying nuclease activity increases gradually from 21 to 25nt bumper region is not supported by the data.

Answer: We thank the reviewer for this concern. In the original **Fig. 2b**, using the fluorescent RNA cleavage assay, we determined the fluorescence signal (normalized to each RNA activator) from each of the resulting time courses. For example, in the section of 21-nt RNA activators with 1-nt bumper, the fluorescence signals of testing groups, such as R1+R2+Bridge, R1+Bridge, R2+Bridge, Bridge and w/o R1, R2, or bridge, were normalized to that of R1+R2+Bridge. The fluorescence data of 23-nt and 25-nt RNA activators were also analyzed in this way. That's the reason why the normalized fluorescence of 21-nt, 23-

nt and 25-nt RNA activators all reached to 1.0 at 60 min. We have reanalyzed the data by normalizing them to the contiguous complementary target RNA (T₂₀) in the updated **Fig. 2b**, which can better show Cas13a nuclease activation gradually improved with the increasing bumper length of RNA activator from 1-nt to 5-nt. We hope we have addressed the confusing points mentioned by the reviewer. The updated **Fig. 2b** has been added in the revised manuscript.

Figure 2b doesn't quite agree with 2c. What is data in Figure 2b normalized to (same question for all other normalized data except for 2c)?

Answer: We thank the reviewer for this concern. The data in **Fig. 2b** have been normalized to the contiguous complementary target RNA (T₂₀) in consistence with **Fig. 2c**, and all normalized data have been explained in their figure legends in the revised manuscript.

Line 188: Figure 2c does not show 50% reduction in the 21nt group compared to the full length.

Answer: The original description in line 188 was '...approximate 50-fold reduction...'. The average apparent cleavage rate of 21-nt RNA activators relative to T₂₀ was 1.94%, while the average apparent cleavage rate of T₂₀ was 96.7%. **Fig. 2c** shows 49.8-fold reduction in the 21-nt group compared to the full length. Therefore, we keep the original description without change.

Line 253: ctrl 2-1 is high. Do the authors have any hypothesis? Is it relevant in a potential therapeutic application?

Answer: We thank the reviewer for this concern. Control 2-1 and Control 2-2 of **Fig. 3b** should have fluorescence signals, both of them can trigger Cas13a HEPN-nuclease catalytic activity, because of the assistance of crRNA' or crRNA as the bridge strand, just like two RNA activators co-activating Cas13a with the assistance of an auxiliary sequence in **Fig. 2**. CRISPR-Cas13a Gemini System has a higher cleavage rate than Control 2-1 and Control 2-2, due to the synergistic trans-RNA cleavage effect of dual Cas13a nucleases.

The quinary complexes of Control 2-1 and Control 2-2 are illustrated in **Supplementary Fig. 10**. The GC

contents of complementary bases between RNA activators and crRNA (or crRNA') in Control 2-1 and Control 2-2 are 60% and 50%, respectively. The higher the GC content, the stronger the binding affinity between nucleic acid strands. The stronger binding of RNA activators:crRNA triplex in Control 2-1 triggers HEPN1 domain to move closer toward HEPN2 domain, activating the HEPN catalytic site of Cas13a nuclease, which subsequently cleaves more RNA reporters in a non-specific manner, comparing to Control 2-2. This may be one reason for the different fluorescence signals between Control 2-1 and Control 2-2. Furthermore, several studies have reported that different sequences of guide region in crRNA will affect Cas13a HEPN-nuclease catalytic activity, resulting in differences in fluorescence signals^{1,2}. We have added **Supplementary Fig. 10** in the supporting information. The associated descriptions and references have been added on **page 16** in the revised manuscript.

Supplementary Fig. 10 Schematic of the quinary complexes of Control 2-1 and Control 2-2.

1. Yang, J. et al. Engineered LwaCas13a with enhanced collateral activity for nucleic acid detection. *Nature Chemical Biology* 19, 45-54 (2023).
2. Tong, H. et al. High-fidelity Cas13 variants for targeted RNA degradation with minimal collateral effects. *Nature Biotechnology* 41, 108-119 (2023).

Line 270: what would happen if the 2 RNA activators are not 1 to 1 ratio?

Answer: We thank the reviewer for this concern. In Fig. 3h, we applied CRISPR-Cas13a Gemini system (Cas13a: crRNA = 1:1, [crRNA] = 1.5 pM) for rapid qualitative diagnosis, like a test strip. According to reviewer's suggestion, two RNA activators with varied ratios (R1: R2 = 1: 10, with [R1] = 1 pM; R1: R2 = 10: 1, with [R1] = 10 pM) have been further studied using this system. The results show that CRISPR-Cas13a Gemini System could significantly distinguish samples containing two RNA targets with varied concentrations (1, 10 and 50 pM) and ratios (1: 10 and 10: 1) from negative control and samples containing only one RNA target ($p < 0.0001$), but there was no significant difference when the concentrations of two RNA targets were not less than 1 pM, regardless of their ratio. These results have been added in **Fig. 3h**, and the associated descriptions have been added on **page 18** in the revised manuscript.

Figure 4C: what are single RNA controls exactly? Are they single crRNA controls? why are these generating strong signals compared to what observed in figure 3 with a testing case? What are some factors that can contribute to this difference between testing cases and real-life application settings?

Answer: We thank the reviewer for this concern. In **Fig. 4c**, the single RNA controls are that traditional CRISPR-Cas13a system (contiguous RNA activation paradigm) tested miR-375 and miR-155, respectively. Controls in **Fig. 3e** were using CRISPR-Cas13a Gemini System which cannot be activated by single target RNA, while single RNA controls in **Fig. 4c** were using traditional CRISPR-Cas13a system, the crRNA of which was designed to be fully complementary to single target RNA for activating Cas13a HEPN-nuclease activity. We hope we have addressed the confusing points mentioned by the reviewer. We have indicated that the single RNA controls were determined by traditional CRISPR-Cas13a system in the caption of **Fig. 4c** and on **page 21** of the revised manuscript.

Reviewer #3 (Remarks to the Author):

Zhao and colleagues present a method named CRISPR-Cas13a Gemini, a method to activate Cas13 cleavage with two different target RNAs that can be harnessed for diagnostics and degrading RNAs in cell

culture. They identify the proper design of crRNAs that creates a loop-like structure that brings together two Cas13a-crRNA complexes and activates cleavage in the presence of two separate target RNAs. They apply the Gemini method for the detection of micro RNAs, Epstein Barr RNAs and cleavage of GFP and RFP simultaneously in transfected cells.

I applaud the authors for the following strengths of their manuscript:

1. The schematics at the start of each figure that nicely depict how the developed is intended to active Cas13 cleavage.
2. Several applications of Gemini to demonstrate its versatility.
3. Inclusion of Fig 1f to demonstrate the potential background of partial crRNA:target binding.

Despite these strengths I have the following concerns:

Major

1. Although Gemini represents a new strategy to activate Cas13 against multiple RNAs, it is unclear from the data and the discussion why this approach is more advantageous than providing two crRNAs that fully match two targets compared to splitting the crRNA complementarity across two RNA targets and thereby requiring a bumper region. In Figure 1, T20 is much more efficient at Cas13 (i.e. faster fluorescence accumulation) than both strategies, so the authors should directly compare in Figure 3 through 5 how Gemini compares to providing two RNA targets one that matches crRNA and one that matches crRNA'. This comparison is critical to understanding the relevance of this approach.

Answer: We thank the reviewer for this concern. 1) Without the assistance of other methods, the traditional CRISPR-Cas13a system (contiguous RNA activation paradigm) can only detect one RNA target in one tube or system. If you want to detect two RNA targets, you must perform them in two tubes or systems, because Cas13a trans-RNA cleavage can cause varying degrees of degradation to all single-stranded RNA molecules in the same system. Comparatively, the advantages of the CRISPR-Cas13a Gemini System include: simultaneous detection of two RNA targets in the same physical space could increase the accuracy of the single readout and deeply explore the correlation between the two RNA targets. In addition, knockdown of

two RNA targets can be achieved at the same time in the same cell. We have added these descriptions in the discussion section of the revised manuscript.

2) **Fig. 1e** reveals the apparent cleavage rates of G₅₋₁₅ to G₁₅₋₅ relative to contiguous target RNA (T₂₀). The data shows that the T₉₊₁₁' resulted in the most robust Cas13a HEPN-nuclease activation, with cleavage rate of 94.6% to T₂₀. We have further verified this result by another set of crRNA' and T'₉₊₁₁' (**Supplementary Fig. 2**). The data show that both T₉₊₁₁' and T'₉₊₁₁' could reach the same fluorescence signal as two contiguous target RNAs (T₂₀ and T'₂₀) at 30 min. As T₉₊₁₁' exhibits the highest Cas13a HEPN-nuclease catalytic activity, it was selected for establishing the CRISPR-Cas13a Gemini System and subsequent application experiments. In **Supplementary Fig. 8**, we have compared CRISPR-Cas13a Gemini System with 21-, 23- or 25-nt RNA activators to traditional CRISPR-Cas13a system with two complementary contiguous targets one that matches crRNA (T₂₀) and one that matches crRNA' (T'₂₀). The data revealed that CRISPR-Cas13a Gemini System with 25-nt RNA activators resulted in similar Cas13a activation relative to traditional CRISPR-Cas13a system with T₂₀ or T'₂₀, whereas, which alone fail to detect two RNAs in a single reaction. In **Fig. 4c**, the single RNA controls were determined by traditional CRISPR-Cas13a systems (contiguous RNA activation paradigm) with crRNAs one that matches miR-375 and one that matches miR-155. Receiver operating characteristic (ROC) curves show that CRISPR-Cas13a Gemini System had higher sensitivity and specificity than single RNA controls, increasing the accuracy of the single readout. We have added these descriptions on **page 10 and 21** in the revised manuscript.

2. Given the data in Figure 1, the authors move to a modification of Strategy II in Figure 2 and use a bridge RNA with a region of the activator that binds the bridge as opposed to using a loop, but a key control is missing in Figure 2b (R1+R2 w/o bridge) as it will directly demonstrate that the extra sequence on the activators doesn't drastically change the efficiency and demonstrate the essential addition of the bridge to activate Cas13 efficiently.

Answer: We thank the reviewer for this valuable suggestion. A control, R1 + R2, has been added in **Fig. 2b**. The data reveal that only two RNA activators (R1+R2) were not able to trigger the Cas13a HEPN-nuclease activity, demonstrating that the extra sequence on the activators doesn't drastically change the

efficiency and the essential addition of the bridge to activate Cas13a efficiently. We have shown the data of **Fig. 2b** by normalizing them to contiguous complementary target RNA (T₂₀). The updated **Fig. 2b** can better show Cas13a nuclease activation gradually improved with the increasing bumper length of RNA activator. We have added the updated **Fig. 2b** in the revised manuscript.

3. It is unclear what is the difference between Gemini and Control 2-1 and Control 2-2. Does it relate to pre-incubation of Cas13 with the crRNAs or activators? Is the concentration of Cas13 different so that there are not as many dual Cas13 complexes? AFM images showing lack of dual peaks would support this difference. Also given the sequence similarities between Control 2-1 and 2-2 the authors should comment on why the background fluorescence is so different.

Answer: We thank the reviewer for this concern and valuable suggestion. In CRISPR-Cas13a Gemini System, Cas13a:crRNA complex and Cas13a:crRNA' complex were preassembled by incubating 150 nM of LbuCas13a with 150 nM of crRNA and crRNA', respectively. Then, the formed two complexes were mixed together and followed by the addition of RNA activators to complete the assembly of the Cas13a:crRNA:RNA activator 1:RNA activator 2:crRNA':Cas13a senary complex. In Control 2-1 (or Control 2-2), Cas13a:crRNA complex (or Cas13a:crRNA' complex) was preassembled by incubating 150 nM of LbuCas13a with 150 nM of crRNA (or crRNA'), followed by the addition of RNA activators and crRNA' (or crRNA). The main difference between CRISPR-Cas13a Gemini System and Control 2-1 and Control 2-2 is that CRISPR-Cas13a Gemini System contains two sets of Cas13a:crRNA complexes, while Control 2-1 and Control 2-2 have only one set of Cas13a:crRNA complex, respectively. As mentioned above, the concentration of Cas13a:crRNA complex in Control 2-1 or Cas13a:crRNA' complex in Control 2-2 was the same as that of the assembled dual Cas13a complex (Cas13a:crRNA:RNA activator 1:RNA activator 2:crRNA':Cas13a senary complex) in CRISPR-Cas13a Gemini System, which was 150 nM.

To better understand the difference in morphology between CRISPR-Cas13a Gemini System and Control 2-1 and Control 2-2, we dissolved the crystal of Control 2-1 (or Control 2-2) and observed it by AFM (**Supplementary Fig. 11**). We have found that many particles containing one set of Cas13a:crRNA complex show single peak morphology, while, the AFM image of CRISPR-Cas13a Gemini System shows a fist-to-fist architecture, demonstrating the formation of dual Cas13a complex (Cas13a:crRNA:RNA activator 1:RNA activator 2:crRNA':Cas13a senary complex) (**Fig. 3d**).

Supplementary Fig. 11 3D and 2D AFM images of Control 2-1 or Control 2-2.

The quinary complexes of Control 2-1 and Control 2-2 are illustrated in **Supplementary Fig. 10**. The GC contents of complementary bases between RNA activators and crRNA (or crRNA') in Control 2-1 and Control 2-2 are 60% and 50%, respectively. The higher the GC content, the stronger the binding affinity between nucleic acid strands. The stronger binding of RNA activators:crRNA triplex in Control 2-1 triggers HEPN1 domain to move closer toward HEPN2 domain, activating the HEPN catalytic site of Cas13a nuclease, which subsequently cleaves more RNA reporters in a non-specific manner, comparing to Control 2-2. This may be one reason for the different fluorescence signals between Control 2-1 and Control 2-2. Furthermore, several studies have reported that different sequences of guide region in crRNA will affect Cas13a HEPN-nuclease catalytic activity, resulting in differences in fluorescence signals^{1,2}. We have added **Supplementary Fig. 10** and **11** in the supporting information. The associated descriptions and references have been added on **page 16, 17** in the revised manuscript.

Supplementary Fig. 10 Schematic of the quinary complexes of Control 2-1 and Control 2-2.

1. Yang, J. et al. Engineered LwaCas13a with enhanced collateral activity for nucleic acid detection. *Nature Chemical Biology* 19, 45-54 (2023).
2. Tong, H. et al. High-fidelity Cas13 variants for targeted RNA degradation with minimal collateral effects. *Nature Biotechnology* 41, 108-119 (2023).

4. Many Cas13 detection methods require pre-amplification e.g. isothermal methods or PCR. The methods do not indicate this was performed for the data presented in Figure 4, but it is unclear what the levels of the

microRNAs and EBERs are in patient samples, and as a result the authors either need to measure this by qPCR or assess the limit of detection of their method. Furthermore, only showing one lateral flow for each sample group is not convincing.

Answer: Thanks for the concern on this issue. The CRISPR-Cas13a Gemini System for dual RNA detection in Fig. 4 was performed without pre-amplification e.g. isothermal methods or PCR. The sample-to-result time of our platform could be less than 60 min, while several hours to days are often needed for RT-PCR, microarray and RNA-Seq. This is a significant advantage over existing techniques in the detection of miRNAs with varying and short half-life¹. The limit of detection of CRISPR-Cas13a Gemini System has been studied in Fig. 3g, which was 733 fM. This fM sensitivity reported here is comparable to that of amplification-based RT-PCR². The levels of the microRNAs (miR-155 and miR-375) in patient serum are in the range of picomolar to nanomolar (Fig. L2)^{3,4}. The EBERs are expressed at high levels (10^7 copies per cell) in cells transformed by EBV⁵. Therefore, the CRISPR-Cas13a Gemini System is capable of detecting these microRNAs and EBERs in patient samples. Moreover, before testing, patient samples were pre-processed by RNA extraction and purification, which could further increase the concentrations of RNA targets. The associated descriptions have been added on page 20, 21 in the revised manuscript.

Table 2 miR-155 detection in clinical samples

Patient ID	[miR155], nM	CV, %
Patient 1	83.5	1.1
Patient 2	35.0	1.6
Patient 3	50.6	1.2
Patient 4	54.7	0.2
Patient 5	58.2	0.7
Healthy	33.4	0.5

Fig. L2 miR-155 (a) and miR-375 (b) expression levels detected in clinical samples.

According to the reviewer's suggestion, we compared the CRISPR-Cas13a Gemini System in fluorescence assay with commercial lateral flow assay (LFA) by testing several clinical samples, both of which

successfully distinguished the positive samples from healthy subjects (**Supplementary Fig. 13**). Notably, CRISPR-Cas13a Gemini System in fluorescence assay and LFA generated consistent signals that high fluorescence signals were corresponding to strong visible signals. Taken together, these results demonstrate the capability of CRISPR-Cas13a Gemini System in the highly sensitive and selective disease screening. We have added **Supplementary Fig. 13** in the supporting information. The associated descriptions and references have been added on **page 20, 21** in the revised manuscript.

Supplementary Fig. 13 Comparison of CRISPR-Cas13a Gemini System in fluorescence assay with commercial LFA in clinical sample detection. a, Fluorescence intensities of CRISPR-Cas13a Gemini System. b, The photos of LFA.

1. Ruegger, S. & Grosshans, H. MicroRNA turnover: when, how, and why. *Trends Biochem. Sci.* 37, 436–446 (2012).
2. Krepelkova, I. et al. Evaluation of miRNA detection methods for the analytical characteristic necessary for clinical utilization. *BioTechniques* 66, 277-284 (2019).
3. Cai, S. et al. Single-molecule amplification-free multiplexed detection of circulating microRNA cancer biomarkers from serum. *Nature Communications* 12, 3515 (2021).

4. Broyles, D., Cissell, K., Kumar, M. & Deo, S. Solution-phase detection of dual microRNA biomarkers in serum. *Analytical and Bioanalytical Chemistry* 402, 543-550 (2012).
5. Khan, G., Coates, P.J., Kangro, H.O. & Slavin, G. Epstein Barr virus (EBV) encoded small RNAs: targets for detection by in situ hybridisation with oligonucleotide probes. *Journal of Clinical Pathology* 45, 616-620 (1992).

5. Figure 4 and 5 are missing key details on the design of the activators. Because of the bridge and crRNA binding requirements do the complementary regions need to be on the end of the RNAs? These design details should be described in the text as they are key to others being able to apply this method to other targets.

Answer: We thank the reviewer for this constructive suggestion. The sequences of the detected RNAs with underlined targeted regions to crRNAs and their corresponding designed crRNAs in **Fig. 4** and **5** have been added in **Supplementary Table 3**.

Besides the strategy I and II in **Fig. 1a**, three additional models have been designed to test the generalizability of our method (**Supplementary Fig. 6**). The length of the RNA sequence used in strategy II was 60-nt. Limited to the current nucleic acid synthesis technology, the purity of synthesized nucleic acid sequence over 100-nt will decrease, therefore, the full length of the synthesized long RNA sequence used in these three models was set at no more than 100-nt.

In model 1A, a short RNA sequence and the target region near 5' end (10, 20, or 40-nt distance from 5' end) of another short RNA sequence together are complementary to the guide region of crRNA. In model 1B, the 3' end and the target region near 5' end (10, 20, or 40-nt distance from 5' end) of a long RNA sequence together are completely complementary to the guide region of crRNA (**Supplementary Fig. 6a**).

In model 2A, a short RNA sequence and the target region near 3' end (10, 20, or 40-nt distance from 3' end) of another short RNA sequence together are complementary to the guide region of crRNA. In model 2B, the 5' end and the target region near 3' end (10, 20, or 40-nt distance from 3' end) of a long RNA sequence together are completely complementary to the guide region of crRNA (**Supplementary Fig. 6c**).

In model 3A, the target region near 5' end (10, 20, or 40-nt distance from 5' end) of a short RNA sequence, and the target region near 3' end (10, 20, or 40-nt distance from 3' end) of another short RNA sequence together are complementary to the guide region of crRNA. In model 3B, the target region near 5' end (10,

20, or 40-nt distance from 5' end), and the target region near 3' end (10, 20, or 40-nt distance from 3' end) of a long RNA sequence together are completely complementary to the guide region of crRNA (**Supplementary Fig. 6e**).

Comparison experiments between short RNA sequences (model 1A, 2A and 3A) and long RNA sequences (model 1B, 2B and 3B) with different locations of target region were performed using G9-11 (**Supplementary Fig. 6b, d, f**). The results show that short RNA sequences in these three models exhibited no Cas13a trans-RNA cleavage regardless of the non-target extension sequences at the 5' and/or 3' end, but Cas13a HEPN-nuclease activity could be triggered when forming a loop constructure with the long RNA sequences regardless of whether the target region is at the end or other locations of the sequence. These additional models demonstrate the spacer sequence of crRNA is not restricted to the 5' and 3' end of any RNA targets, indicating the broad generalizability of our method. **Supplementary Fig. 6** and **Table 3** has been added in the supplementary information, and the associated description has been added on **page 8, 20** in the revised manuscript.

Supplementary Fig. 6 Noncontiguous target RNA activation paradigm of Cas13a. **a**, Schematic of model 1 for activating Cas13a. **b**, Apparent cleavage rate of model 1 for Cas13a trans-cleavage relative to T_{20} . **c**, Schematic of model 2 for activating Cas13a. **d**, Apparent cleavage rate of model 2 for Cas13a trans-cleavage relative to T_{20} . **e**, Schematic of model 3 for activating Cas13a. **f**, Apparent cleavage rate of model 3 for Cas13a trans-cleavage relative to T_{20} .

Minor

1. Throughout the paper the normalization of the data should be explained in the figure legends.

Answer: Thanks for this suggestion. The normalization of the data has been explained in each figure legend in the revised manuscript.

2. Figure 1D: Legend is more easily interpreted if ordered sequentially by strategy I followed by strategy II and adding SI and SII labels because the subscripts are subtle.

Answer: We thank the reviewer for this suggestion. The legend in **Fig. 1d** has been rewritten to improve the clarity in the revised manuscript.

3. Figure 2b: Legend should be R1, R2, and bridge (not or)

Answer: Thanks for this suggestion. We have made this correction in **Fig. 2b** in the revised manuscript.

4. Line 370-371 claim: reference?

Answer: We thank the reviewer for this reminder. The associated references have been cited in the revised manuscript.

5. The authors should be more cautious throughout when proposing that their method detects two RNA targets (in the case of miRNAs and EBERs) because the way it is phrased it could be misinterpreted to be a multiplexed diagnostic that differentially detects two targets, but in fact it is that Cas13 is activated by

two targets increasing the sensitivity of the single readout.

Answer: Thanks for this reminder. We have been more cautious throughout and revised the way it is phrased that our method detects two RNA targets in the revised manuscript.

REVIEWERS' COMMENTS

Reviewer #1 (Remarks to the Author):

The author did an excellent job addressing my concerns. I do not have further comment. Good job.

Reviewer #3 (Remarks to the Author):

As with the initial submission, I was intrigued by this new method for activating Cas13 with a loop-like structure. My initial major concern regarding how Gemini would compare to a system in which two full length crRNAs are provided (unlinked) was likely due to confusion of how detection of two targets was defined/described and what the intended goal of Gemini was.

Upon initially reading, I felt that the goal of this method was to generate a positive result if either of the two target RNAs were present. For example, in the case of breast cancer diagnosis, you might want to detect a positive case if either miR-155 and miR-375 were present and in that case two full length crRNAs would be sufficient. If the goal is to only have a positive signal when both are present together, then yes providing two full length RNAs would not provide the intended result. If this was the author's intent, then Gemini as an approach is essential and powerful.

With this better understanding and the edits, I am pleased with the revised manuscript. The authors have included the necessary controls and additional experiments that satisfied both my major and minor concerns. My one remaining suggestion is to be even more explicit about the utility of this non-contiguous approach in the introduction and the discussion. It is important to make it clear that the method will only produce a signal if both RNAs are present not one or the other and as mentioned above I think in certain instances this can be quite powerful.

In the following, we present our response (marked in blue) to each review comment (marked in yellow) in detail.

REVIEWERS' COMMENTS

Reviewer #1 (Remarks to the Author):

The author did an excellent job addressing my concerns. I do not have further comment. Good job.

Answer: We thank the reviewer for his/her kind approval.

Reviewer #3 (Remarks to the Author):

As with the initial submission, I was intrigued by this new method for activating Cas13 with a loop-like structure. My initial major concern regarding how Gemini would compare to a system in which two full length crRNAs are provided (unlinked) was likely due to confusion of how detection of two targets was defined/described and what the intended goal of Gemini was.

Upon initially reading, I felt that the goal of this method was to generate a positive result if either of the two target RNAs were present. For example, in the case of breast cancer diagnosis, you might want to detect a positive case if either miR-155 and miR-375 were present and in that case two full length crRNAs would be sufficient. If the goal is to only have a positive signal when both are present together, then yes providing two full length RNAs would not provide the intended result. If this was the author's intent, then Gemini as an approach is essential and powerful.

With this better understanding and the edits, I am pleased with the revised manuscript. The authors have included the necessary controls and additional experiments that satisfied both my major and minor concerns.

My one remaining suggestion is to be even more explicit about the utility of this non-contiguous approach in the introduction and the discussion. It is important to make it clear that the method will only produce a signal if both RNAs are present not one or the other and as mentioned above I think in certain instances this can be quite powerful.

Answer: We thank the reviewer for his/her kind approval. According to this suggestion, we have added

some statements about the utility of this non-contiguous approach in the introduction and the discussion of the revised manuscript.